# Machine learning predictor PSPire screens for phase-separating proteins lacking intrinsically disordered regions

Shuang Hou [1,6], Jiaojiao Hu [2,3,6], Zhaowei Yu[1], Dan Li [4,5], Cong Liu [2,3] ✉ & Yong Zhang [1] ✉

The burgeoning comprehension of protein phase separation (PS) has ushered in a wealth of bioinformatics tools for the prediction of phase-separating proteins (PSPs). These tools often skew towards PSPs with a high content of intrinsically disordered regions (IDRs), thus frequently undervaluing potential PSPs without IDRs. Nonetheless, PS is not only steered by IDRs but also by the structured modular domains and interactions that aren't necessarily reflected in amino acid sequences. In this work, we introduce PSPire, a machine learning predictor that incorporates both residue-level and structure-level features for the precise prediction of PSPs. Compared to current PSP predictors, PSPire shows a notable improvement in identifying PSPs without IDRs, which underscores the crucial role of non-IDR, structure-based characteristics in multivalent interactions throughout the PS process. Additionally, our biological validation experiments substantiate the predictive capacity of PSPire, with 9 out of 11 chosen candidate PSPs confirmed to form condensates within cells.

The intricate regulation of complex biochemical reactions within cells has always been an essential issue. Membrane-bound organelles, surrounded by phospholipid bilayers, physically separate their interior and exterior environments, ensuring a stable reaction environment. However, membraneless organelles (MLOs), such as nucleoli and stress granules, can concentrate proteins and nucleic acids at specific cellular sites without bounding membranes. The formation, composition control, and function regulation of these MLOs have been elusive for years. In 2009, a study found that P granules in germ cells from *Caenorhabditis elegans* can form liquid-like droplets, suggesting phase separation (PS) could underlie the formation of these biomolecular condensates[1]. Subsequent studies implicated PS in various

fundamental biological processes like transmembrane signaling[2], DNA repair[3], transcription[4,5], and RNA processing[6]. Abnormal formation or disruption of biomolecular condensates can cause neurodegenerative disorders[7], cancer[8,9], and infectious diseases[10].

A key feature of phase-separating proteins (PSPs) is their capacity to form multiple weak, transient, noncovalent interactions. The stickers-and-spacers model offers an intuitive perspective to the driving-force behind PS[11,12]. In this model, the stickers represent protein-protein or protein-RNA interaction domains, while the spacers are interspersed between stickers and can modulate PS behavior[13]. A considerable number of PSPs can form biomolecular condensates via interactions between intrinsically disordered regions (IDRs), which

[1]State Key Laboratory of Cardiology and Medical Innovation Center, Institute for Regenerative Medicine, Department of Neurosurgery, Shanghai East Hospital, Shanghai Key Laboratory of Signaling and Disease Research, Frontier Science Center for Stem Cell Research, School of Life Sciences and Technology, Tongji University, Shanghai 200092, China. [2]Interdisciplinary Research Center on Biology and Chemistry, Shanghai Institute of Organic Chemistry, Chinese Academy of Sciences, Shanghai 201210, China. [3]State Key Laboratory of Chemical Biology, Shanghai Institute of Organic Chemistry, Chinese Academy of Sciences, Shanghai 200032, China. [4]Bio-X Institutes, Key Laboratory for the Genetics of Developmental and Neuropsychiatric Disorders, Ministry of Education, Shanghai Jiao Tong University, Shanghai 200240, China. [5]Zhangjiang Institute for Advanced Study, Shanghai Jiao Tong University, Shanghai 200240, China. [6]These authors contributed equally: Shuang Hou, Jiaojiao Hu. ✉e-mail: liulab@sioc.ac.cn; yzhang@tongji.edu.cn

possess highly flexible conformations and present multiple weakly interacting elements. IDRs typically contain more charged and polar amino acids, while often lacking bulky hydrophobic amino acids necessary for forming well-structured domains. In addition to IDRs, another way to achieve multivalent interactions is through modular interaction domains. These domains serve as binding modules and contribute significantly to PS by oligomerizing multiple protein molecules, effectively increasing the number of interaction sites and the multivalency. In this study, we categorized PSPs into two groups: those containing IDRs (ID-PSPs) and those without IDRs (noID-PSPs). The IDRs were determined based on pLDDT scores from AlphaFold-predicted protein structures (see Methods for details).

The development of computational methods for predicting PSPs is crucial for facilitating the rapid *in-silico* screening of the entire proteome. The initial PSP prediction models focused on specific or limited protein sequence features, utilizing only small subsets of the entire proteome[14]. PLAAC[15], catGRANULE[16], PScore[17], PSPer[18], and several other methods[19–21] belong to this category. In recent years, with the surge in PSP studies[22], more comprehensive PSP prediction methods have been developed, such as FuzDrop[23], PSAP[24], PSPredictor[25], and PhaSePred[26]. These recent methods outperformed their predecessors, mainly due to the usage of larger training datasets and the implementation of machine-learning techniques. Despite these advantages, current PSP predictors severely biased towards predicting ID-PSPs, resulting in subpar performances in predicting noID-PSPs (see Results for details). This bias underscores the prevailing challenge of accurately identifying PSPs without IDRs.

As the structures of noID-PSPs may offer insights into the multivalent interactions underlying their functions, we hypothesize that incorporating protein structural information could significantly enhance the prediction of noID-PSPs. Current PSP predictors rely solely on amino acid sequences and do not leverage protein structural information, likely due to the limited availability of high-quality protein structures. Recently, AlphaFold emerged as the top-performing method for predicting 3D protein structures with near-experimental accuracy[27], and the AlphaFold Protein Structure Database made high-accuracy structure predictions publicly accessible[28].

In this work, leveraging the availability of high-accuracy atomic coordinates of proteins in the full human proteome, we train an XGBoost classifier, PSPire, to predict PSPs by incorporating both residue-level and structure-level features. We employ the PS-related features utilized for the prediction of PSPs by the two best current predictors, PSAP and PhaSePred, and calculate these features on IDRs and non-IDRs separately. Evaluations using various datasets demonstrate that our model significantly outperforms current predictors in classifying noID-PSPs from non-PSPs, highlighting the significant value of protein structural information in decoding the multivalency involved in PS.

## Results

### Current PSPs predictors struggle to accurately predict noID-PSPs

We initially gathered human PSPs from two sources: (1) PSPs employed in the development of PhaSePred[26]; (2) PSPs extracted from LLPSDB[29], PhaSePro[30], PhaSepDB[31], and DrLLPS[32] databases. The PS abilities of these PSPs were corroborated by in vivo or in vitro experiments or by the identification of membraneless compartments. Subsequently, these PSPs were randomly divided into training and testing datasets (see Methods for details). As evidence suggested that several initial PSP prediction models exhibit heavy bias towards proteins with high IDR contents[33], we hypothesized that the current PSP predictors might be less effective in predicting noID-PSPs. To verify our suspicion, we assessed the performances of representative PSP predictors (including PhaSePred, PSPredictor, PSAP, FuzDrop, PSPer, PScore, catGRANULE, and PLAAC; Supplementary Table 1) on separate ID-PSPs

and noID-PSPs datasets (see Methods for details). The area under the receiver operating characteristic curve (AUROC) and the area under the precision-recall curve (AUPRC) revealed that these predictors were quite effective in distinguishing ID-PSPs from non-PSPs (best AUROC = 0.84, best AUPRC = 0.42). However, their ability to predict noID-PSPs was significantly lower (best AUROC = 0.68, best AUPRC = 0.08) (Fig. 1a, Supplementary Fig. 1a), highlighting the ongoing challenge of accurately identifying PSPs that do not contain IDRs.

To understand why current predictors were less effective in predicting noID-PSPs, we investigated the features employed in PSAP and PhaSePred, which are considered the best performers among current PSP predictors. PSAP utilized a set of elaborately designed amino acid features associated with PS, out of which we analyzed the 10 most impactful ones by comparing their values among ID-PSPs, noID-PSPs, and non-PSPs in our dataset. Interestingly, only the "fraction_L" feature (i.e., the fraction of leucine in the whole protein sequence) was able to simultaneously distinguish ID-PSPs and noID-PSPs from non-PSPs. Seven features demonstrated opposing tendencies for ID-PSPs and noID-PSPs. For example, the "IDR_50" feature (i.e., IDR percentage with an IDR score cutoff of 0.5) showed higher values for ID-PSPs compared to non-PSPs and lower values for noID-PSPs compared to non-PSPs. The remaining two features ("fraction_C", i.e., the fraction of cysteine in the whole protein sequence, and "group_Xle", i.e., the fraction of Xle group which includes leucine and isoleucine) did not show statistically significant difference between noID-PSPs and non-PSPs (Fig. 1b). Similarly, when examining the six features used by PhaSePred, comparable trends were observed (Supplementary Fig. 1b). Only two features ("phos_frequency", i.e., the phosphorylation frequency, and "group_Charged", i.e., the fraction of charged group which includes K, R, D, and E) was able to simultaneously distinguish ID-PSPs and noID-PSPs from non-PSPs. Taken together, it appears that most features employed by current PSP predictors do not favor noID-PSPs, which explains their subpar performance in predicting noID-PSPs. This observation underscores the need for utilizing other features suitable for the accurate prediction of noID-PSPs.

### Structured superficial features enable identification of both ID-PSPs and noID-PSPs

Considering that noID-PSPs lack IDRs, we aimed to identify IDR-independent features that can effectively distinguish both ID-PSPs and noID-PSPs from non-PSPs. As each residue in a protein can either be buried within the protein structure or exposed to the surrounding solvent, and that exposed residues are often implicated in interactions with other proteins or ligands, we hypothesized that residues of structured superficial regions (SSUP) of PSPs might play a significant role in the multivalency involved in PS. To explore this hypothesis, we identified SSUP for each protein in our dataset by using AlphaFold-predicted 3D structures (Fig. 2a; see Methods for details). Among the most impactful features of PSAP and PhaSePred (Fig. 1b, Supplementary Fig. 1b), eight of them could be calculated based on residues of SSUP. As the definition of SSUP intrinsically excludes IDRs, these features derived from SSUP residues can be considered IDR-independent. Surprisingly, in contrast to the unfavorable full-length protein features for noID-PSPs, all of the SSUP features could concurrently differentiate ID-PSPs and noID-PSPs from non-PSPs (Fig. 2b). For example, the "group_Hydrophobic" feature (i.e., the fraction of hydrophobic group which includes V, I, L, M, F, W and Y) of the full-length protein displayed opposing tendencies for ID-PSPs and noID-PSPs (Fig. 1b), whereas the same feature derived from SSUP demonstrated similar tendencies for ID-PSPs and noID-PSPs (Fig. 2b). This observation emphasized that SSUP residues directly contribute to PS, and features derived from SSUP could be utilized for the accurate prediction of both ID-PSPs and noID-PSPs.

To further elucidate the role of SSUP residues in PS, we explored the stickers-and-spacers model in the specific context of SSUP.

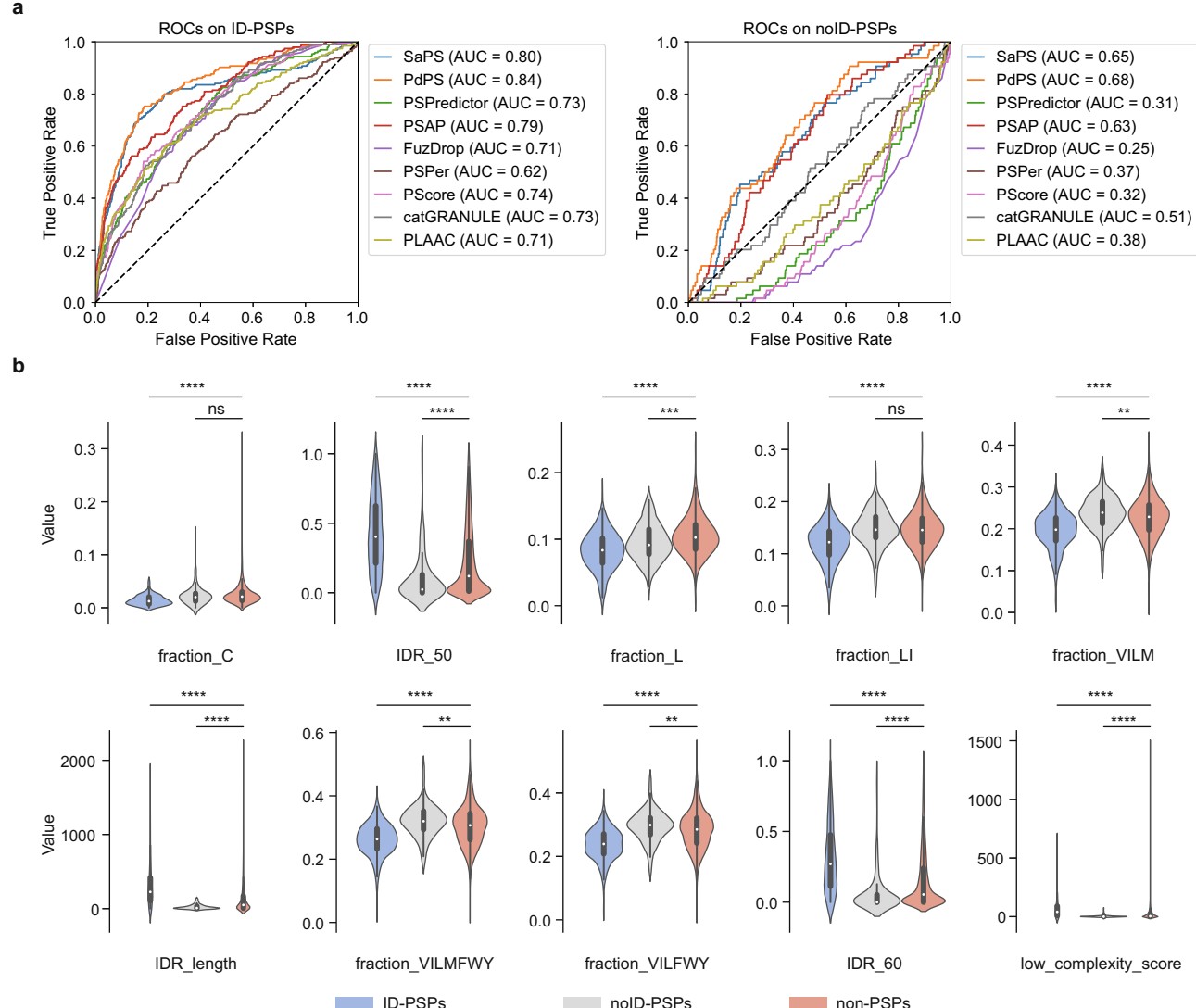

**Fig. 1 | Performance of current PSP predictors on ID-PSPs and noID-PSPs.**
**a** Receiver operating characteristic curves (ROC) of eight predictors on testing dataset. The performance was evaluated on ID-PSPs and noID-PSPs separately. The PhaSePred tool includes two models: SaPS for self-assembling proteins and PdPS for partner-dependent proteins. **b** Comparison of ten PS-related features that attribute the highest importance to PSAP prediction between the two types of PSPs (ID-PSPs and noID-PSPs) and non-PSPs. The amino acid groups include Xle (L, I), Aliphatic (V, I, L, M), Hydrophobic (V, I, L, M, F, W, Y), and Alpha helix (V, I, L, F, W, Y). The ten features calculated on the whole protein sequence are: fraction_C (i.e., the fraction of cysteine), fraction_L (i.e., the fraction of leucine), IDR_50 (i.e., IDR percentage with an IDR score cutoff of 0.5), IDR_60 (i.e., IDR percentage with an IDR score cutoff of 0.6), IDR_length (i.e., IDR length with an IDR score cutoff of 0.5), the proportion of the four amino acid groups, and low complexity score. *P* values were calculated using the two-sided Mann-Whitney U test: ns (not significant) for $p > 0.05$, ** for $p < 0.01$, *** for $p < 0.001$, and **** for $p < 0.0001$. The central dot indicates the median. The box represents interquartile range (IQR), 25–75th percentile. Whiskers extend to the data's minima and maxima within 1.5 * IQR. The comparison was conducted on the union of training and testing datasets which contained 389 ID-PSPs, 128 noID-PSPs, and 10,284 non-PSPs.

Considering that electrostatic interactions between positively and negatively charged residues are well-studied bases for sticker interactions contributing to PS behavior, we designed a computational approach for the identification of charged stickers, i.e., clusters of similarly charged residues interspersed on SSUP (Fig. 2c; see Methods for details). For our approach, we tested a range of distances from 10 Å to 20 Å and chose 14 Å as the threshold where the fraction of proteins with ≥3 stickers in the union of training and testing datasets reached the maximum (Supplementary Fig. 2a; see Methods for details). As PS has been reported to be mediated by electrostatic interactions through the stickers in the coiled-coil domain[34], LINE-1 ORF1 protein was shown as an example of calculated stickers in the coiled-coil domain (Fig. 2d). By applying our approach, we identified stickers for each protein in the union dataset. We observed that the proportion of proteins with ≥3 stickers was higher in both ID-PSPs and noID-PSPs compared to non-PSPs (Supplementary Fig. 2b). Moreover, considering that the presence of both positively and negatively charged stickers could promote the formation of electrostatic interactions between proteins, we further calculated the number of sticker pairs for each protein, i.e., the minimum number of the positively and negatively charged stickers. We found that the proportion of proteins with ≥2 sticker pairs was higher in ID-PSPs and noID-PSPs than in non-PSPs (Supplementary Fig. 2c). To eliminate the differences in the number of SSUP residues, we calculated normalized values for sticker number and sticker pair number, and compared these values among ID-PSPs, noID-PSPs, and non-PSPs, revealing that both normalized values for ID-PSPs and noID-PSPs were significantly higher than those of non-PSPs (Fig. 2e). These results supported that SSUP residues have the potential to mediate PS via the stickers-and-spacers model.

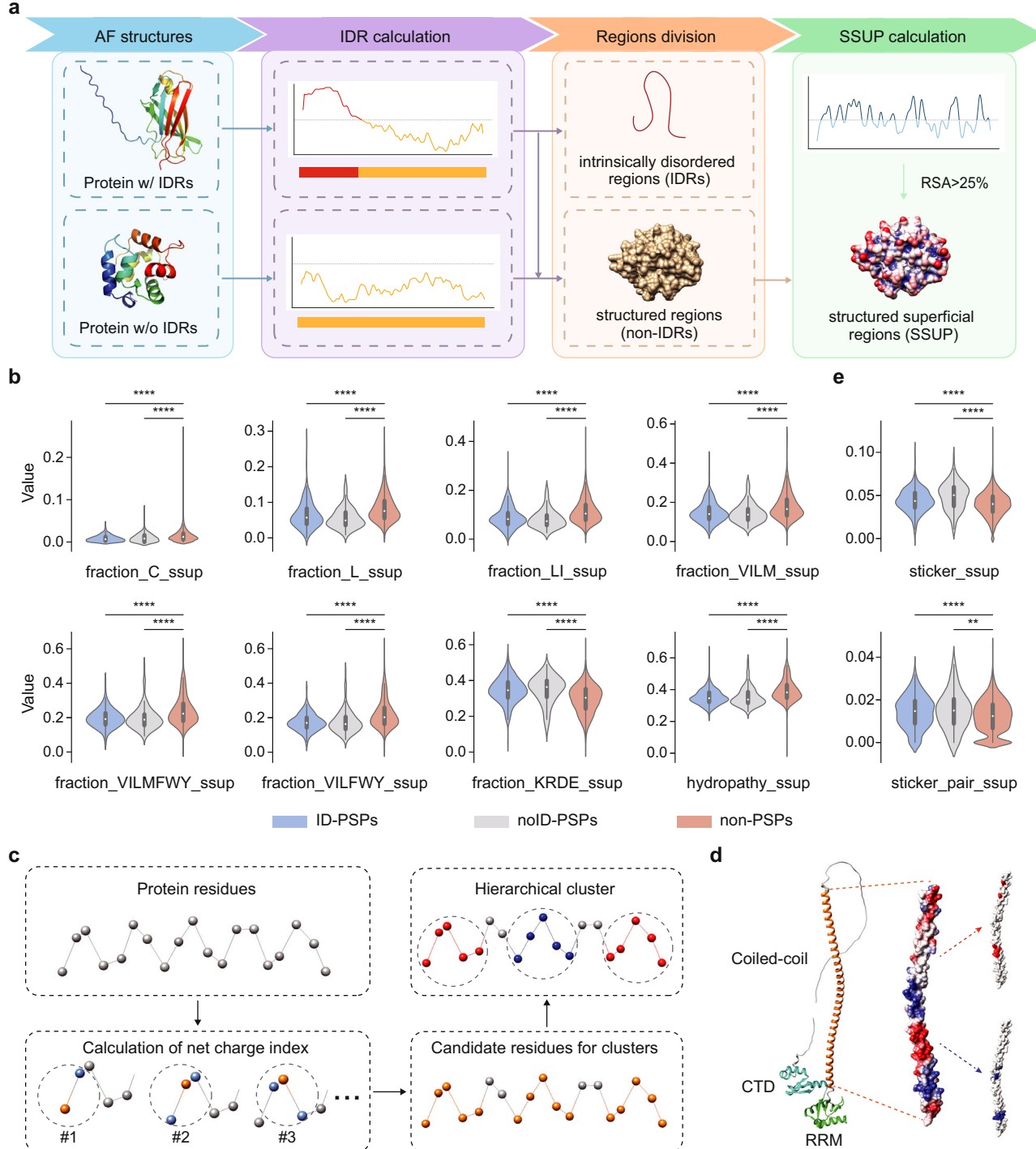

## Development and performance evaluation of PSPire

In order to leverage the distinct features observed on SSUP for both ID-PSPs and noID-PSPs compared to non-PSPs, we aimed to develop a high-accuracy machine learning classifier that could effectively distinguish both ID-PSPs and noID-PSPs from non-PSPs by integrating these features. In addition to the aforementioned top features from PSAP and PhaSePred, we incorporated all other PS-correlated features employed by PSAP. These additional features were first calculated on SSUP as were done for the aforementioned features. Besides SSUP features, we also calculated all features on IDRs since PS of many ID-PSPs is dependent on IDRs, and these IDR-related features were null for proteins without IDRs. Furthermore, we also incorporated the phosphorylation (Phos) frequency feature, using Phos sites of human

proteins from PhosphoSitePlus[35], which has been demonstrated to be IDR-independent and harbored the leading contribution to the prediction of PhaSePred[26]. By incorporating the IDR- and SSUP-related features along with the Phos frequency feature (Supplementary Data 1), we designed an XGBoost predictor of PSPs, named PSPire, based on a combination of residue-level and structure-level features to predict PS propensity for both ID-PSPs and noID-PSPs (Fig. 3 and Supplementary Fig. 3; see Methods for details). As Phos sites recorded in PhosphoSitePlus are sparse for species other than human, we trained models with the Phos feature for predicting PSPs in human, while trained models without the Phos feature for predicting PSPs for other species. We utilized the model interpreter SHAP separately to distinguish ID-PSPs and noID-PSPs from non-PSPs in the human testing

**Fig. 2 | Features calculated on structured superficial regions (SSUP).**
**a** Schematic view of SSUP calculation. Intrinsically disordered regions (IDRs) of a protein were first determined based on AlphaFold (AF) structures. The protein was then divided into IDRs and non-IDRs. Lastly, the residues in the non-IDRs with a relative solvent accessibility (RSA) value greater than 25% constituted the SSUP.
**b, e** Comparison of ten SSUP-related features between the two types of PSPs (ID-PSPs and noID-PSPs) and non-PSPs. The amino acid groups include Xle (L, I), Ali-phatic (V, I, L, M), Hydrophobic (V, I, L, M, F, W, Y), Alpha helix (V, I, L, F, W, Y), and Charged (K, R, D, E). The eight features calculated on SSUP are: (**b**) fraction_C_ssup (i.e., the fraction of cysteine), fraction_L_ssup (i.e., the fraction of leucine), the proportion of the five amino acid groups, hydropathy_ssup (i.e., hydropathy score), (**e**) sticker_ssup (i.e., normalized value of charged sticker number), and stick-er_pair_ssup (i.e., normalized value of charged sticker pair number). *P* values were calculated using the two-sided Mann-Whitney U test: ** for $p < 0.01$ and **** for $p < 0.0001$. The central dot indicates the median. The box represents interquartile

range (IQR), 25–75th percentile. Whiskers extend to the data's minima and maxima within 1.5 * IQR. The comparison was conducted on the union of training and testing datasets which contained 389 ID-PSPs, 128 noID-PSPs, and 10,284 non-PSPs.
**c** Schematic view of stickers calculation. First, the net charge index of each residue in SSUP within a defined distance was calculated. Then, a group of residues with an absolute value of the net charge index greater than three were collected. Lastly, hierarchical clustering was performed on the group of residues to obtain positive and negative clusters. **d** Graphical representation of charged stickers in SSUP. The example shown is the LINE-1 ORF1 protein. The left panel displays the 3D structure of three domains: the coiled-coil domain, the RNA recognition motif (RRM), and the C-terminal domain (CTD). The middle panel presents the protein surface of the coiled-coil domain colored by Coulombic electrostatic potential which ranges from negative (red) to positive (blue). The right panel shows the calculated stickers (red for negative stickers and blue for positive stickers) of the coiled-coil domain using our algorithm.

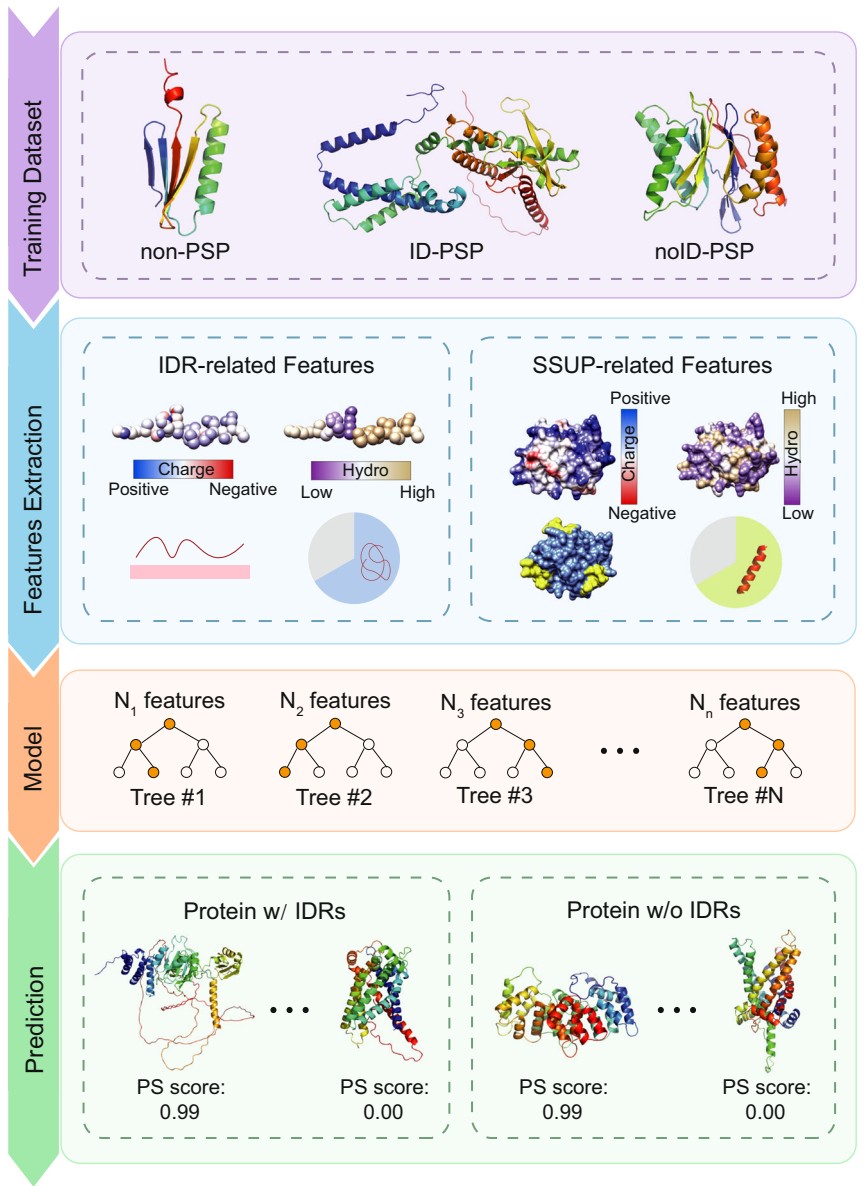

**Fig. 3 | Schematic workflow of PSPire.** The training dataset contained three types of proteins: two types of PSPs (ID-PSPs and noID-PSPs) and non-PSPs. IDR- and SSUP-related features were calculated for all proteins in the training dataset. An XGBoost classifier was then constructed using these features. For a query protein, PSPire can output the PS score which denotes the likelihood of phase separation and the protein type which indicates whether the protein contains IDRs.

dataset (see Methods for details) to measure the contribution of each feature. The Phos frequency feature got the highest absolute SHAP score in both cases. Besides the Phos feature, IDR- and SSUP-related features were both important for ID-PSPs prediction, while the top-ranked features for noID-PSPs prediction were all calculated on SSUP as noID-PSPs do not contain IDRs (Supplementary Fig. 4).

We evaluated the performance of PSPire in predicting noID-PSPs and ID-PSPs on the human testing dataset. The performance of PSPire for ID-PSPs prediction (AUROC: 0.86, AUPRC: 0.51; Fig. 4a, b) was comparable to the top current predictors (AUROC: 0.84, AUPRC: 0.42; Fig. 1a, Supplementary Fig. 1a). Notably, PSPire demonstrated superior performance for noID-PSPs prediction (AUROC: 0.84, AUPRC: 0.24; Fig. 4a, b) in contrast to the best current predictors (AUROC: 0.68, AUPRC: 0.08; Fig. 1a, Supplementary Fig. 1a). Given the scarcity of in vivo or in vitro confirmed PSPs, we further evaluated the performance of PSPire using 5 human datasets annotating proteome in MLOs, since

proteins in MLOs are potential PSPs (see Methods for details). As shown in Fig. 4c, PSPire performed comparably to the best current predictors in predicting ID-PSPs across all five datasets, yet it demonstrated significantly better performance for noID-PSPs prediction. In addition to AUROC and AUPRC, other evaluation metrics (i.e., Matthews correlation coefficient (MCC), F1-score, sensitivity, specificity, accuracy, false positive rate (FPR), and false negative rate (FNR)) showed similar trends on the testing dataset and the five MLO datasets (Supplementary Table 2). In summary, the results showed that PSPire remarkably outperforms current predictors in distinguishing noID-PSPs from non-PSPs.

**Validation of candidate PSPs predicted by PSPire**
To systematically predict candidate human PSPs, we ultimately trained PSPire using the combination of training and testing datasets and utilized it to assign the PS scores to all human proteins (Supplementary

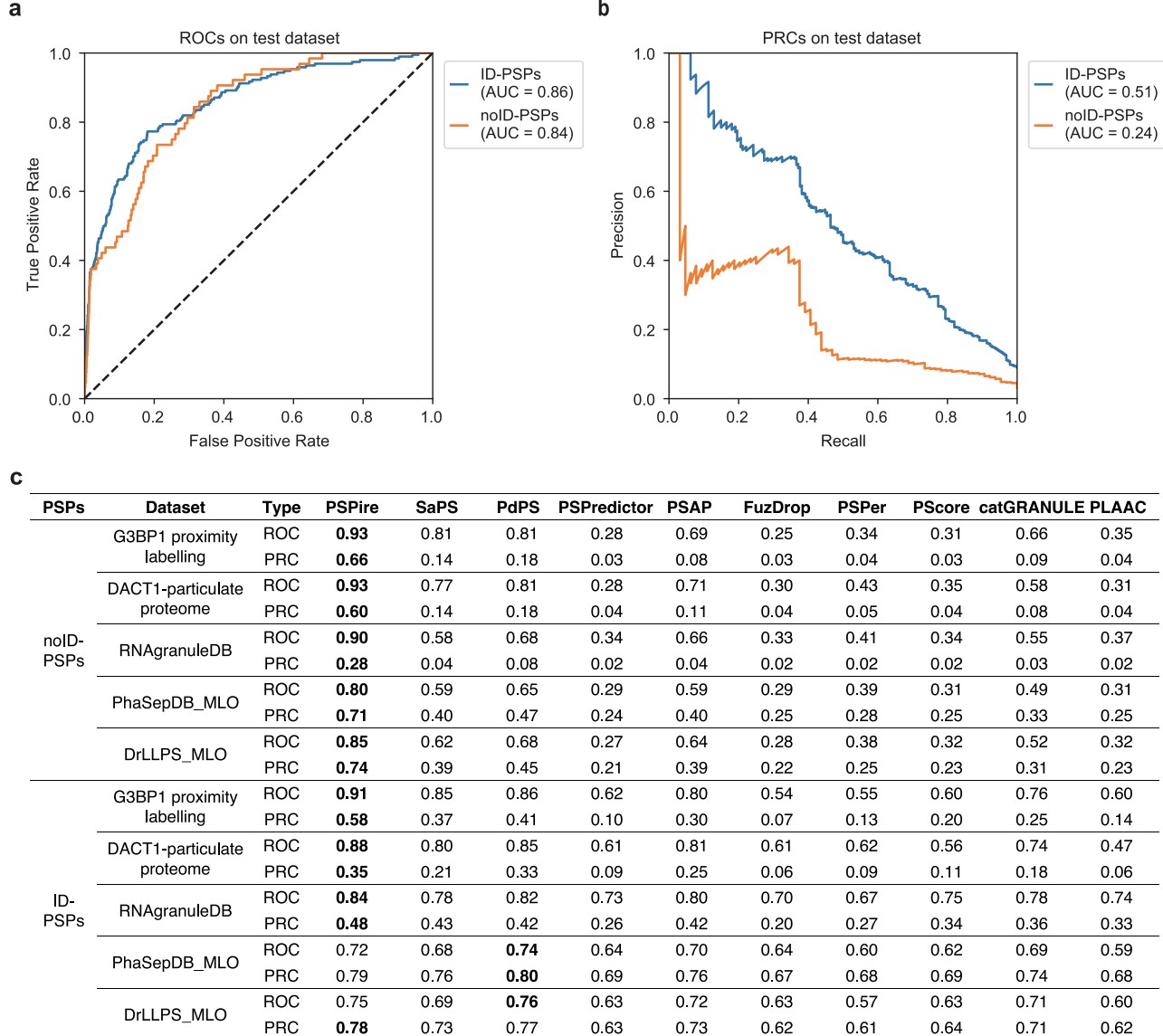

**c**

| PSPs | Dataset | Type | PSPire | SaPS | PdPS | PSPredictor | PSAP | FuzDrop | PSPer | PScore | catGRANULE | PLAAC |
|---|---|---|---|---|---|---|---|---|---|---|---|---|
| noID-PSPs | G3BP1 proximity labelling | ROC | **0.93** | 0.81 | 0.81 | 0.28 | 0.69 | 0.25 | 0.34 | 0.31 | 0.66 | 0.35 |
| | | PRC | **0.66** | 0.14 | 0.18 | 0.03 | 0.08 | 0.03 | 0.04 | 0.03 | 0.09 | 0.04 |
| | DACT1-particulate proteome | ROC | **0.93** | 0.77 | 0.81 | 0.28 | 0.71 | 0.30 | 0.43 | 0.35 | 0.58 | 0.31 |
| | | PRC | **0.60** | 0.14 | 0.18 | 0.04 | 0.11 | 0.04 | 0.05 | 0.04 | 0.08 | 0.04 |
| | RNAgranuleDB | ROC | **0.90** | 0.58 | 0.68 | 0.34 | 0.66 | 0.33 | 0.41 | 0.34 | 0.55 | 0.37 |
| | | PRC | **0.28** | 0.04 | 0.08 | 0.02 | 0.04 | 0.02 | 0.02 | 0.02 | 0.03 | 0.02 |
| | PhaSepDB_MLO | ROC | **0.80** | 0.59 | 0.65 | 0.29 | 0.59 | 0.29 | 0.39 | 0.31 | 0.49 | 0.31 |
| | | PRC | **0.71** | 0.40 | 0.47 | 0.24 | 0.40 | 0.25 | 0.28 | 0.25 | 0.33 | 0.25 |
| | DrLLPS_MLO | ROC | **0.85** | 0.62 | 0.68 | 0.27 | 0.64 | 0.28 | 0.38 | 0.32 | 0.52 | 0.32 |
| | | PRC | **0.74** | 0.39 | 0.45 | 0.21 | 0.39 | 0.22 | 0.25 | 0.23 | 0.31 | 0.23 |
| ID-PSPs | G3BP1 proximity labelling | ROC | **0.91** | 0.85 | 0.86 | 0.62 | 0.80 | 0.54 | 0.55 | 0.60 | 0.76 | 0.60 |
| | | PRC | **0.58** | 0.37 | 0.41 | 0.10 | 0.30 | 0.07 | 0.13 | 0.20 | 0.25 | 0.14 |
| | DACT1-particulate proteome | ROC | **0.88** | 0.80 | 0.85 | 0.61 | 0.81 | 0.61 | 0.62 | 0.56 | 0.74 | 0.47 |
| | | PRC | **0.35** | 0.21 | 0.33 | 0.09 | 0.25 | 0.06 | 0.09 | 0.11 | 0.18 | 0.06 |
| | RNAgranuleDB | ROC | **0.84** | 0.78 | 0.82 | 0.73 | 0.80 | 0.70 | 0.67 | 0.75 | 0.78 | 0.74 |
| | | PRC | **0.48** | 0.43 | 0.42 | 0.26 | 0.42 | 0.20 | 0.27 | 0.34 | 0.36 | 0.33 |
| | PhaSepDB_MLO | ROC | 0.72 | 0.68 | **0.74** | 0.64 | 0.70 | 0.64 | 0.60 | 0.62 | 0.69 | 0.59 |
| | | PRC | 0.79 | 0.76 | **0.80** | 0.69 | 0.76 | 0.67 | 0.68 | 0.69 | 0.74 | 0.68 |
| | DrLLPS_MLO | ROC | 0.75 | 0.69 | **0.76** | 0.63 | 0.72 | 0.63 | 0.57 | 0.63 | 0.71 | 0.60 |
| | | PRC | **0.78** | 0.73 | 0.77 | 0.63 | 0.73 | 0.62 | 0.61 | 0.64 | 0.71 | 0.62 |

**Fig. 4 | Performance benchmarking of PSPire against current PSP predictors.** **a**, **b** Performance of PSPire on the testing dataset assessed by ROC and PRC curves. The performance was evaluated on ID-PSPs and noID-PSPs separately. **c** AUCs of ROC and PRC for PSPire and eight predictors on five human MLO datasets: the G3BP1 proximity labeling set, the DACT1-particulate proteome set, the RNA-granuleDB Tier1 set, the PhaSepDB low and high throughput MLO set, and the DrLLPS MLO set. The PhaSePred tool includes two models: SaPS for self-assembling proteins and PdPS for partner-dependent proteins. AUC values are calculated by using ID-PSPs or noID-PSPs in these datasets as positive samples and proteins in the negative testing dataset as negative samples. Proteins in the positive training dataset were excluded. The best results for each row are marked in bold.

Data 2). The proteins were then ranked based on their PS scores, and those ranked within the top 1500 but not present in the training dataset or MLOs were considered as highly confident PSP candidates, whereas those ranked between the top 1500 and 3500 were termed as moderately confident candidates. In total, 74 highly confident ID-PSP candidates, 100 highly confident noID-PSP candidates, 580 moderately confident ID-PSP candidates, and 395 moderately confident noID-PSP candidates were identified (Supplementary Data 3). Among the PSP candidates, 152 proteins were predicted as candidate PSPs by DeepPhase[36], a predictor based on immunofluorescence images from the Human Protein Atlas. Besides, orthologs of 11 PSP candidates in other species have been recognized as PSPs, which were validated by in vivo or in vitro experiments. Additionally, orthologs of 99 other PSP candidates have been reported as MLO compartments in different species. Recently, 11 ID-PSP candidates and 5 noID-PSP candidates were reported to exhibit the ability to undergo PS. These evidences suggested the reliability of candidates predicted by PSPire.

To substantiate the reliability of PSPire's predictions, we conducted further biological experiments to authenticate the PS propensity of the candidate PSPs. In addition to the PS score, we considered other factors in selecting candidates, such as the protein size, literature support of association with the biological processes of interest, the availability of antibodies for immunofluorescence, and the feasibility of expressing and purifying proteins for in vitro condensation assays. We first chose three highly confident noID-PSP candidates (HPRT1, H3C15, and ACTR2) and three negative candidates (ZNF738, GTF2A2, and CA5B) for validation. AlphaFold-predicted 3D structures of the three noID-PSP candidates can be seen in Supplementary Fig. 5. We constructed GFP-tagged versions of these proteins and overexpressed them in HeLa cells. As shown in Fig. 5a, GFP-ACTR2, GFP-H3C15, and GFP-HPRT1 obviously formed condensates in cytoplasm or nuclei. In contrast, cells transfected with GFP-ZNF738, GFP-GTF2A2, or GFP-CA5B showed negligible puncta both in cytoplasm and nuclei (Fig. 5b).

We further conducted validation studies for several candidates at the endogenous expression level. Given the commercial availability of primary antibodies, we focused on two highly confident noID-PSP candidates (ANXA3 and S100A7) and three moderately confident candidates (PGM1, TXNL4B, and SERPINB4). Additionally, we selected three ID-PSP candidates (CKMT2, RAB31, and VPS26B), which were predicted as noID-PSPs by ESpritz[37] and MobiDB-lite[38], with PS scores akin to the highly confident candidates for CKMT2 and moderately confident candidates for RAB31 and VPS26B when IDR-related features were null. AlphaFold-predicted 3D structures of the eight chosen proteins can be seen in Supplementary Fig. 5. To gauge cellular localization of these proteins under standard conditions, we conducted immunostaining of these endogenous proteins in HeLa cells. PGM1, TXNL4B, SERPINB4, and VPS26B emerged as small cytoplasmic puncta within cells (Fig. 5c). Interestingly, these four proteins' condensates co-localized with EDC4, a known P-body marker, hinting at their role in P-body assembly (Fig. 5c and Supplementary Fig. 6a). S100A7 and RAB31, although dispersed in HeLa cell cytoplasm (Supplementary Fig. 6b), co-localized with cytoplasmic stress granules under sodium arsenite-induced stress (Fig. 5d and Supplementary Fig. 6a). No puncta were observed for ANXA3 and CKMT2 (Supplementary Fig. 6c, d).

These cellular findings affirm that 9 out of 11 selected candidates can form condensates within cells. To validate these proteins' independent in vitro PS, we obtained purified full-length proteins of PGM1, SERPINB4, S100A7, and RAB31 (Supplementary Fig. 6e). Introducing crowding agent PEG 3350 led to the formation of typical spherical liquid droplets by PGM1, SERPINB4, and RAB31 - a key PSP trait. Similarly, S100A7 underwent PS in the presence of another crowding agent, PEG 8000 (Fig. 5e). We expanded our investigation by labeling PGM1, SERPINB4, S100A7, and RAB31 with fluorescent dyes and conducting

fluorescence recovery after photobleaching (FRAP) experiments. The FRAP results revealed moderate dynamics in the condensates formed by PGM1, RAB31, and S100A7, and relatively limited dynamics in SERPINB4 condensates, as detailed in Supplementary Fig. 6f–i. Given that PSPire incorporates the features of charged stickers, we further examined electrostatic interactions among them in our candidates. SERPINB4 and PGM1, with high counts of charged stickers (10 and 11, respectively), were postulated to form condensates driven by electrostatic interactions. High salt concentrations indeed significantly disrupted the phase separation of SERPINB4 and PGM1 (Supplementary Fig. 6j). Collectively, these results verify these proteins' strong PS propensity, reinforcing the efficacy of PSPire.

## Discussion

The burgeoning understanding of proteins and their biological functions through the formation of biomolecular condensates emphasizes the importance of accurate PSP predictors. PSP predictors could allow researchers to identify PSP candidates from the proteome, thereby expediting our comprehension of the PS process. In this study, we developed PSPire, a machine learning model developed to predict PS propensities based on the integration of residue-level and structure-level features. Unlike current predictors that primarily rely on amino acid features, PSPire integrates 3D structural information, demonstrating superior performance in identifying noID-PSPs. Consequently, PSPire effectively identified PSP candidates and could benefit our understanding of these proteins and their role in condensate formation.

Multivalent interactions driving phase separation not only involve IDR-driven nonspecific interactions but also widely concern modular domain-mediated specific interactions[39,40]. However, most existing PSP predictors displayed a marked bias towards proteins with high IDR contents, resulting in suboptimal performance when predicting noID-PSPs. To address this, we introduced non-IDR features based on SSUP to complement IDR-related features. Our analyses showed that these SSUP-related features effectively distinguish PSPs from non-PSPs, indicating a strong correlation between SSUP residues and the multivalency inherent to the PS process of structural domain-driving proteins. Furthermore, we computed sticker-related features that could differentiate PSPs from non-PSPs effectively. Hence, SSUP residues, particularly those constituting stickers, offer sites where mutations might impact PS behavior, which could be valuable for further experimental validation and have the potential to aid in the identification of drug targets related to PS. Besides biological experiments, critical residues in SSUP can be further explored using molecular dynamics to uncover potential mechanisms driving PS. Leveraging these important features, PSPire reported the residue positions of SSUP and identified stickers as output.

The theoretical framework, known as the stickers-and-spacers model, describes the molecular grammar underlying various phase-separating systems (Supplementary Fig. 7). These systems can be categorized into three distinct types: folded proteins, intrinsically disordered proteins, and linear multivalent proteins[41]. For folded proteins, stickers are defined as interaction patches on the protein surface, while spacers consist of regions that do not engage in interactions. In intrinsically disordered proteins, stickers may include individual amino acids, short linear motifs, or a combination of both, interspersed by spacers, which are the intervening non-interactive residues. Regarding linear multivalent proteins, stickers comprise the multiple folded domains, and spacers are the flexible linkers that connect these domains. For well-defined binding domains, stickers are characterized as the binding sites on the domain surfaces, with non-binding surface residues serving as additional spacers. From another perspective, the computed features related to stickers, IDRs, and SSUP were designed to capture the distinct properties of the three types of stickers accurately.

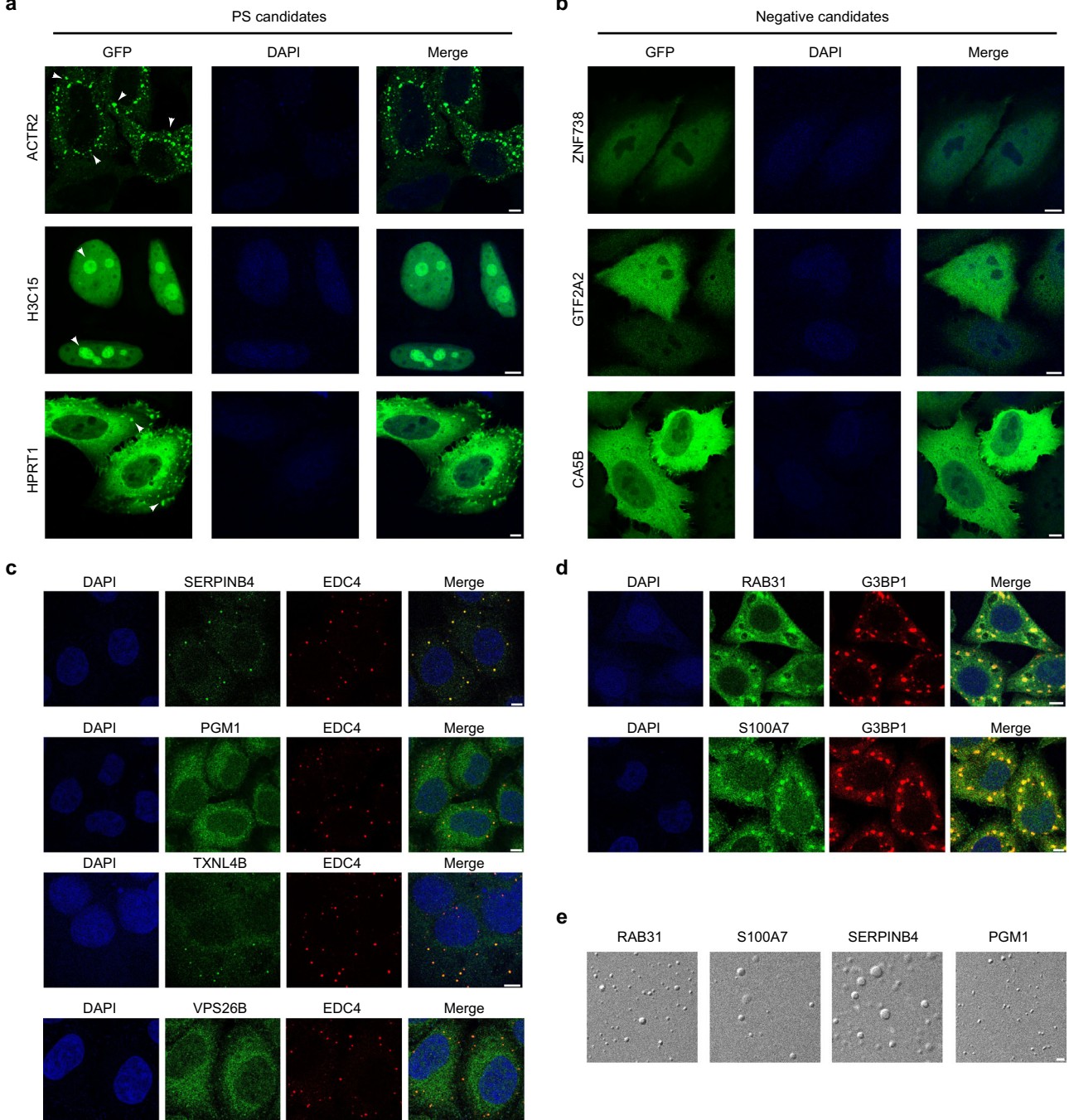

**Fig. 5 | Candidate PSPs predicted by PSPire undergo phase separation in HeLa cells and in vitro. a** Confocal images of GFP-ACTR2, GFP-H3C15, and GFP-HPRT1 overexpressing in HeLa cells. Condensates were indicated by white arrows. Scale bars, 5 μm. **b** Confocal images of GFP-ZNF738, GFP-GTF2A2, and GFP-CA5B expressing in HeLa cells. Scale bars, 5 μm. **c** Confocal images of endogenous PGM1, TXNL4B, SERPINB4, and VPS26B in HeLa cells. EDC4 (red) marks the P-body. Scale bars, 5 μm. **d** Immunostaining images of endogenous S100A7 and RAB31 in HeLa cells under stress conditions induced by sodium arsenite. G3BP1 (red) marks the stress granules. Scale bars, 5 μm. **e** Differential interference contrast (DIC) images demonstrate PS of PGM1, SERPINB4, RAB31, and S100A7 under specified conditions. Scale bars, 2 μm. The imaging was independently repeated 3 times with similar observations.

Regarding sticker-related features, PSPire focused on electrostatic interactions, instead of hydrophobic interactions, with the following considerations. The strength of electrostatic interactions, ranging from 2 to 15 kcal/mol, is typically greater than that of hydrophobic interactions, which range from 0.5 to 3 kcal/mol[42]. In addition, the proportions of hydrophobic residues in SSUP were significantly lower in ID-PSPs and noID-PSPs than in non-PSPs, while the proportions of charged residues in SSUP were significantly higher in ID-PSPs

and noID-PSPs compared to non-PSPs (Fig. 2b), suggesting that electrostatic interactions might be more prevalent than hydrophobic interactions for structural domain-driven phase separation. We attempted to modify the sticker identification method by incorporating hydrophobic residues. However, the incorporation of hydrophobic residues made no improvement in the prediction power of PSPire.

While PSPire's performance is exemplary, it does face several challenges. First, despite AlphaFold's near-experimental precision in

predicting 3D protein structures from amino acid sequences[43], discrepancies can arise between the AlphaFold-predicted and native structures. For instance, AlphaFold predicts the structure of α-Syn to comprise an N-terminal α-helix domain and a C-terminal IDR, whereas its native structure has three unstructured domains. Given that protein structures significantly contribute to calculating structured superficial regions, the performance of PSPire hinges on the accuracy of its predicted structure. We expect enhancements in prediction performance as more precise protein structures become available. Alternatively, users could provide their own refined protein structures for enhanced accuracy in predictions. Second, the current classification between ID-PSPs and noID-PSPs is based on the presence of IDRs. Nevertheless, the PS of certain ID-PSPs predominantly hinges on modular domains rather than IDRs. A case in point is JAK1, a recognized ID-PSP, which can form a PS-driven CNF1-JAK1-JAK2 complex via its SH2 domain[44]. With PSPire's default features, the PS score of JAK1 is a mere 0.64. To tackle the issue that IDR-related features may inadvertently impair prediction, PSPire offers a parameter to specifically ignore IDR-related features for proteins with IDRs. Consequently, when JAK1's IDR-related features are nullified, its PS score increases to 0.93. Third, PSPire is not designed to indicate which proteins are more likely to be drivers or passengers of phase separation, mainly due to the composition of the training dataset. Further experiments, such as the in vitro purified protein condensation assays, are required to confirm the proteins' role as drivers or passengers and capabilities to form either single- or multi-component droplets within cells.

## Methods

### Datasets
In this study, datasets used for development of the PhaSePred[26] model were obtained, which contained 155 PSPs and 8801 non-PSPs for training, and 117 PSPs and 2200 non-PSPs for testing. Then a total of 189 additional PSPs were retrieved from LLPSDB[29], PhaSePro[30], PhaSepDB[31], and DrLLPS[32], which were validated by in vivo or in vitro experiments. Besides, 77 noID-PSPs classified as scaffolds or regulators by DrLLPS were included. Since proteins longer than 2700 amino acids were segmented into overlapping fragments by AlphaFold, proteins with a sequence length ≤100 or ≥2700 amino acids were filtered out. As PSPer[18], PScore[17], and PLAAC[15] have restrictions on the length of protein sequences, proteins that cannot be predicted by these three tools were also filtered out. Then the ID-PSPs and noID-PSPs were both randomly split into separate training and testing datasets with a ratio of 1:1 and the training PSPs of PhaSePred were reserved in training dataset. Consequently, the positive training dataset comprised 259 PSPs, including 195 ID-PSPs and 64 noID-PSPs; the positive testing dataset consisted of 258 PSPs with 194 ID-PSPs and 64 noID-PSPs; and the negative training and testing datasets contained 8323 and 1961 proteins, respectively (Supplementary Data 4). The union dataset of training and testing datasets was used for features comparison of ID-PSPs and noID-PSPs from non-PSPs. Furthermore, five human MLO datasets were collected for evaluation: the G3BP1 proximity labeling set[26,45], the DACT1-particulate proteome set[26,46], the RNAgranuleDB Tier1 set[47], the PhaSepDB low and high throughput MLO set[31], and the DrLLPS MLO set[32] (Supplementary Data 5).

### Secondary structure calculation
The secondary structure state of each residue was calculated by the Definition of Secondary Structure of Proteins (DSSP)[48,49] using Alpha-Fold Protein Data Bank (PDB) coordinate files. The DSSP module from the Biopython package was used as an interface to the DSSP program. The resulting secondary structure assignments contained eight types: α-helix (H), 3-helix (G), 5-helix (I), β-bridge (B), β ladder (E), bend (S), turn (T), and irregular. These types were further grouped into three categories: helix (H, G, and I), sheet (B and E), and loop (S, T, and irregular).

### Intrinsically disordered regions (IDRs) calculation
The amino acid sequence and pLDDT scores of each protein were extracted from PDB files of AlphaFold-predicted protein structures using the PDB and SeqIO modules from the Biopython package. To identify long disordered regions, IDRs were assigned based on pLDDT scores using a threshold of 50 (i.e., a residue with a score lower than 50 was considered as disordered). Then residues annotated as helix or sheet secondary structures by the DSSP program[48,49] were filtered out from IDRs. As carried out by MobiDB-lite[38], the IDRs were further refined by iteratively converting short stretches of up to three residues of IDRs among ordered regions to order, and vice versa. Ordered stretches of up to 10 consecutive residues were then converted to IDRs if they were flanked by two IDRs of at least 20 residues. Finally, IDRs with a sequence length below 20 were removed. The final IDRs were used to distinguish between ID-PSPs and noID-PSPs, i.e., PSPs with IDRs were classified as ID-PSPs and PSPs without IDRs as noID-PSPs.

### Structural superficial regions (SSUP) determination
Relative solvent accessible surface area (RSA) is defined as the per-residue ratio between solvent accessible surface area (SASA) and the 'standard' value for a particular residue. In this study, we used the PSAIA program[50] to calculate RSA. First, we grouped the residues based on a threshold value of RSA. If the RSA percentage of a residue was greater than 25%, it was assigned as an exposed residue; otherwise, it was classified as a buried residue. Further, all exposed residues were classified as the superficial group (SUP group), and the non-IDRs were regarded as structural group (S group). Finally, the structural superficial regions (SSUP) were generated by overlapping the S group and SUP group.

### Sticker-related features
To obtain sticker-related features, we first calculated the net charge index of each residue in SSUP. The net charge index of a residue was defined as the number of positive residues in SSUP minus the number of negative residues in SSUP within a defined distance of the residue. We tested a range of distances from 10 Å to 20 Å. The proximate residues were searched using the PyMOL python package as an interface to the PyMOL software[51]. We then obtained a group of residues whose absolute value of the net charge index was greater than three. Next, we implemented the hierarchical clustering of the group of residues using the Python scipy package based on Cα 3D coordinates of each residue, which were extracted from AlphaFold PDB files. The parameters used were: criterion=distance, metric=Euclidean, and method=centroid. The distance thresholds of hierarchical clustering were used as the same as proximate residues searching. Further, if the net charge index of the majority of residues was positive (negative), the cluster was classified as a positive (negative) cluster. Finally, the total sticker number was defined as the sum of the positive and negative cluster numbers, while the sticker pair number was defined as the minimum value of the positive and negative cluster numbers. The final two features of sticker frequency and sticker pair frequency were defined as the total sticker number and sticker pair number divided by the residue number in SSUP.

### IDR- and SSUP-related features
Firstly, amino acids were classified into fifteen groups according to different properties. The combinations included Asx (D, N), Glx (E, Q), Xle (I, L), Positively charged (K, R, H), Negatively charged (D, E), Aromatic (F, W, Y, H), Aliphatic (V, I, L, M), Small (P, G, A, S), Hydrophilic (S, T, H, N, Q, E, D, K, R), Hydrophobic (V, I, L, F, W, Y, M), Alpha helix (V, I, Y, F, W, L), Beta turn (N, P, G, S), Beta sheet (E, M, A, L), Aromaticity (F, W, Y), and Charged (K, R, D, E). The hydropathy value was allocated to each residue, which was calculated using the same method as in localCIDER[52] based on a normalized Kyte-Doolittle hydrophobicity scale[53]. Besides, the polarity value[54] was also allocated to each residue.

Subsequently, the following features, utilized by PSAP and PhaSePred, were calculated on IDRs and SSUP separately: fraction of each of the 20 standard amino acids, proportion of each of the fifteen groups, averaged hydropathy score, as well as the isoelectric point and molecular weight determined by ProteinAnalysis module from the Biopython package. Additionally, the averaged polarity scores of residues in IDRs and SSUP were computed separately and included as features. Besides, the sequence length of IDRs and sequence length percentage of IDRs in a protein were added. Lastly, the phosphorylation (Phos) frequency feature was calculated using the same definition as PhaSePred, i.e., the number of Phos sites retrieved from PhosphoSitePlus[35] divided by the length of the protein sequence.

## Model training and performance evaluation

The feature data was normalized by scaling using the MinMaxScaler from Scikit-learn. We utilized a tree-based ensemble learning method, XGBoost, to train the classifier for distinguishing between PSPs and non-PSPs. During the XGBoost model fitting process, sample weights were assigned to ID-PSPs, noID-PSPs, and non-PSPs in the training dataset based on the inverse of frequencies of each type to reduce the effect of sample imbalance. Two separate models with and without the Phos frequency feature were trained. To mitigate overfitting and enhance the model's generalization capability, we implemented a five-fold cross-validation scheme and utilized the Optuna[55] framework to systematically tune the following hyperparameters: learning_rate, n_estimators, max_depth, min_child_weight, subsample, colsample_bytree, gamma, reg_lambda, and reg_alpha (Supplementary Table 3). The optimization process was guided by the objective of maximizing the area under the receiver operating characteristic curve (AUROC) averaged on the five-fold for prediction of ID-PSPs. A comprehensive set of 1000 trials was executed to ensure a thorough exploration of the hyperparameter space. The optimal set of hyperparameters identified from the best-performing trial was then adopted for the final model (Supplementary Table 4).

For each fold of the five-fold cross-validation process, the positive training, positive validation, and negative validation subsets remained fixed and ten models were generated with ten different negative training subsets. These negative subsets were randomly sampled from the negative training dataset, ensuring that the number of proteins in the negative training subsets was twice the number of proteins in the positive training subsets. Furthermore, the ratio of ID-PSPs to noID-PSPs was consistently maintained across all folds. The final prediction score for a protein was determined by averaging the scores across these ten models. Utilizing the optimal set of hyperparameters, a final training model was trained with the full training dataset for subsequent comparison with existing PSP predictors. Similarly, ten rounds of training were performed, each with a distinct negative training subset, again ensuring a two-to-one ratio of negative to positive proteins. The mean prediction score of the ten rounds was used as the final prediction score. To facilitate a fair comparison, the PSAP model was re-trained on the same training dataset with the same strategy.

The prediction was evaluated using the independent testing dataset. The model's performance was assessed by several metrics: the area under the curve (AUC) of the receiver operating characteristic (ROC) curve and the precision-recall (PRC) curve, Matthews correlation coefficient (MCC), F1-score, sensitivity, specificity, accuracy, false positive rate (FPR), and false negative rate (FNR). To calculate MCC, F1-score, sensitivity, specificity, accuracy, FPR, and FNR, prediction scores were converted to binary labels using a threshold from the point nearest to the top-left corner of the ROC curve. To get more reliable PS scores of the human proteome, a final model was trained with merged training and testing datasets. The same ten-round training procedure was applied to the merged datasets, and the averaged prediction score of the ten trained models was used as the PS score for each protein. A random seed of 42 was used consistently throughout the process to ensure reproducibility.

## Plasmids and constructs

The genes of full length ACTR2, H3C15, HPRT1, ZNF738, GTF2A2, CA5B, RAB31, S100A7, SERPINB4, and PGM1 were synthesized by AZENTA. These genes were inserted into the pCAG vector, which contained an N-terminal GFP tag. For protein purification, RAB31, S100A7, SERPINB4, and PGM1 genes were inserted into the pET-28a vector, which contained an N-terminal His$_6$-tag and a thrombin cleavage site.

## Cell cultures, transfection, and immunofluorescence

HeLa cells obtained from cell bank of the Chinese Academy of Science, Shanghai, China (SCSP-504) were cultured in Dulbecco's Modified Eagle Medium (11995073, Gibco) supplemented with 10% (v/v) fetal bovine serum (10099141, Gibco) and 1% penicillin/streptomycin (15140122, Gibco) at 37 °C in 5% CO$_2$. Transient transfection was performed using Lipofectamine 3000 (Invitrogen) in Opti-MEM (Invitrogen). Cells were transfected for at least 24 h before the subsequent drug treatments or examinations. For immunostaining, cells were grown on coverslips in a 24-well plate. After being washed with PBS, the cells were fixed in 4% paraformaldehyde for 15 min at room temperature. Following fixation, the cells were permeabilized with 0.5% Triton X-100 in PBS for 15 min and blocked with 3% goat serum in PBST (0.1% Triton X-100 in PBS) for 30 min. Next, the cells were incubated with primary antibodies overnight at 4 °C, followed by the incubation with secondary antibodies at room temperature for 1 h. After being washed three times with PBST, the cells were mounted on glass slides using the antifade mountant with DAPI (P36962, Thermo Fisher). For sodium arsenite treatment, cells were incubated with a culture medium containing 250 μM sodium arsenite for 1 h before harvesting. Imaging was performed using a Leica TCS SP8 microscope with a 100 × objective (oil immersion, NA = 1.4) at room temperature. Image J (v 2.0.3) was applied for data processing.

The following antibodies were used for immunofluorescence assays: rabbit anti-PGM1 (abs117064, absin), rabbit anti-SERPINB4 (abs134793, absin), rabbit anti-S100A7 (abs139303, absin), rabbit anti-TXNL4B (abs117670, absin), rabbit anti-RAB31 (abs134703, absin), rabbit anti-VPS26B (absin134908, absin), mouse anti-G3BP1 (611127, BD Biosciences,), mouse anti-EDC4 (sc-376382, Santa Cruz Biotechnology). The following fluorescent secondary antibodies were used: goat anti-rabbit-Alexa Flour 488 (Invitrogen, A-11008), and goat anti-mouse-Alexa Flour 568 (Invitrogen, A-11004). Antibodies including Anti-PGM1, anti-SERPINB4, anti-S100A7, anti-TXNL4B, anti-RAB31, and anti-VPS26B were diluted to a concentration ratio of 1:100. Anti-G3BP1 and anti-EDC4 antibodies were diluted at a higher concentration ratio of 1:500. Secondary antibodies were diluted to a concentration ratio of 1:1000.

## Protein expression and purification

The S100A7, SERPINB4, and PGM1 plasmids were transformed into BL21 (DE3) Chemically Competent Cell (CD601-03, TransGenBiotech). As for the RAB31 plasmid, the Transetta (DE3) Chemically Competent Cell (CD801-02, TransGenBiotech) was used. Cells were grown to an OD$_{600}$ of 0.8 and induced with 0.5 mM IPTG overnight at 16 °C. Proteins were loaded onto the HisTrap FF column (GE Healthcare) with buffer containing 50 mM Tris-HCl, pH 8.0, 500 mM NaCl, and 10% glycerol. The proteins were eluted with imidazole and then further purified using a size exclusion column. The RAB31, S100A7, and SERPINB4 proteins were purified using a Superdex 75 16/600 column (GE Healthcare), while the PGM1 protein was purified using a Superdex 200 16/600 column (GE Healthcare). The purified proteins were stored in a buffer containing 50 mM Tris-HCl, pH 8.0, 500 mM NaCl, and 10% glycerol at −80 °C. The purity of recombined proteins was characterized by SDS-PAGE following the data processing by Image Lab (v 3.0).

For fluorescence labeling, Alexa-488 (A10254, Invitrogen, USA) was used for RAB31, S100A7, SERPINB4, and PGM1. All the labeling experiments were performed as described by the manufacturer. The 2-fold dyes were mixed with proteins and incubated at room temperature for at least 1 h. After that, the proteins labeled with fluorescent dyes were purified using a Superdex 75 Increase 10/300 GL column (GE Healthcare).

## Differential interference contrast imaging

For DIC observation, the purified proteins were diluted in a buffer containing 50 mM Tris and 150 mM NaCl at pH 7.5, to achieve a final concentration of 50 μM. Although a 2% PEG 20000 concentration is often standard[56], optimal conditions indeed require empirical determination for each protein system under investigation. For example, 10% PEG 8000 was used for the PS of TIA1[57], lactoferrin[58], β-lactoglobulin[58], lysozyme[58], RNase A[58], and Tau[58]. Additionally, 15% PEG 8000 was used for the PS of alpha-synuclein[59], TIP60[60], and SOD1[61]. Here, 15% (w/v) PEG 3350 was used for phase separation (PS) of RAB31, PGM1, and SERPINB4, and 10% (w/v) PEG 8000 was used for the PS of S100A7. Once PS was induced in the tube, 3 μL of the solution was pipetted onto a glass slide for DIC imaging. The images were collected using a Leica TCS SP8 microscope with a 100x objective (oil immersion, NA = 1.4) at room temperature.

## Fluorescence recovery after photobleaching (FRAP) assay

The FRAP assay was executed using the FRAP module on a Leica TCS SP8 confocal microscope, equipped with a 100× oil immersion objective. This procedure involved selectively bleaching fluorescently labeled assemblies with a laser beam and targeting a specific circular region of interest. Following photobleaching, imaging was performed continuously, capturing one frame every 2.58 s. The fluorescence intensity in the bleached region (Itm) was measured, along with the intensity (Itc) in a nearby unbleached assembly serving as a control. For quantitative analysis, the fluorescence intensity at the bleached site at each time point (t) was normalized against the control. The recovery of fluorescence was calculated using the formula: It = (Itm/I0m)/(Itc/I0c). All captured images were subsequently analyzed using the Leica Application Suite X software.

## Visualization and statistical analysis

All plotting and statistical analyses were implemented in Python with numpy and pandas packages or R. When applicable, multiple test corrections were carried out using the Benjamini-Hochberg (BH) correction method. $P$ values were calculated using the two-sided Mann-Whitney U test and were indicated in figures as follows: ns (not significant) for $p > 0.05$, * for $p < 0.05$, ** for $p < 0.01$, *** for $p < 0.001$, and **** for $p < 0.0001$. The graphical representations of proteins were generated using PyMOL[51] or UCSF Chimera[62] software.

## Reporting summary

Further information on research design is available in the Nature Portfolio Reporting Summary linked to this article.

## Data availability

The PDB format files of human protein structures (identifier: UP000005640) can be downloaded from the AlphaFold DB website (https://ftp.ebi.ac.uk/pub/databases/alphafold/latest/UP000005640_9606_HUMAN_v4.tar). All datasets used in this study are publicly available and detailed in Supplementary Data 4 and Supplementary Data 5. The following four databases were used for the collection of phase-separating protein datasets: LLPSDB, PhaSePro (https://phasepro.elte.hu), PhaSepDB (http://db.phasep.pro), and DrLLPS (https://llps.biocuckoo.cn). Pre-calculated PSPire predicted scores and residue positions in structured superficial regions (SSUP) and sticker regions for proteins in the following model organism proteomes

generated in this study have been deposited in the GitHub repository (https://github.com/TongjiZhanglab/PSPire): *Arabidopsis thaliana*, *Caenorhabditis elegans*, *Candida albicans*, *Danio rerio*, *Dictyostelium discoideum*, *Drosophila melanogaster*, *Escherichia coli*, *Glycine max*, *Homo sapiens*, *Methanocaldococcus jannaschii*, *Mus musculus*, *Oryza sativa*, *Rattus norvegicus*, *Saccharomyces cerevisiae*, *Schizosaccharomyces pombe*, and *Zea mays*. The secondary structure states and relative surface exposure data of proteins in human proteome generated in this study could also be downloaded from the GitHub repository. Source data are provided with this paper.

## Code availability

PSPire is freely available at https://github.com/TongjiZhanglab/PSPire.

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

## Acknowledgements

We thank Guohui Chuai for his helpful advice. This work was supported by the National Natural Science Foundation of China (32030022, 32325012, 31970642, 82188101, 92353302, and 32170683), the National Key Research and Development Program of China (2021YFA1302500), the Science and Technology Commission of Shanghai Municipality

(23JS1401200, 2019SHZDZX02, and 22JC1410400), the CAS Project for Young Scientists in Basic Research (Grant No. YSBR-095), and the Shanghai Pilot Program for Basic Research – Chinese Academy of Science, Shanghai Branch (Grant No. CYJ-SHFY-2022-005).

## Author contributions

Y.Z. conceived the project; S.H. performed method development and data analysis with the help of Z.Y.; J.H. performed experimental verification under the instruction of C.L. and D.L.; S.H., Y.Z., J.H. and C.L. wrote the manuscript.

## Competing interests

The authors declare no competing interests.
