## [Peer Review File · Nature Communications]

Reviewers' Comments:

Reviewer #1:

Remarks to the Author:

Review of PSPire: a machine learning predictor for high-performance screening of phase-separating proteins without intrinsically disordered regions, by Shuang Hou, Jiaojiao Hu, Zhaowei Yu, Cong Liu, and Yong Zhang. Submitted for consideration of publication in Nature Communications.

This manuscript by Hou & Hu et al., sets out to create a better computational predictor of phase separating proteins, particularly for those that do not contain significant disordered content. In the field, emphasis has been placed on intrinsically disordered domains for driving biomolecular condensation in living cells, which has translated into emphasis in the features utilized for training predictors of potential phase separating proteins. However, many examples of phase separating proteins exist that do not require or utilize disordered regions for phase separation. In this study, Hou & Hu et al. created an alternate metric for predicting self-interaction propensity ("stickers" in the stickers-and-spacers model) that works for well-folded proteins in addition to disordered proteins. They employ alpha fold to identify folded and disordered regions, which can be modified and updated as alpha fold itself progresses. They then choose 8 candidate proteins predicted to have high phase separation potential from their prediction algorithm, and find that 6 of these 8 localize to punctate structures in living cells and demix from aqueous solution with a PEG crowder in vitro, so they conclude this supports their capability of phase separation.

The premise of this manuscript is broadly interesting for audiences in the fields of computational biology, biomolecular condensation and protein structure prediction. I commend the authors for building the PSPire tool, which represents an advance in our ability to comprehensively predict phase separating proteins, and will be useful for hypothesis generation in future studies.

The weaknesses of the manuscript lie in communicating the details of their training and testing data sets, and in demonstrating that the predicted phase-separating proteins indeed are capable of condensation. See below for suggestions on how to strengthen these points. Overall, I recommend publication with minor edits.

Major comments:

- Please include a sentence or two in the main text describing how the authors created the training and testing datasets; in this draft these important parameters are only written in the methods. Specifically, it was unclear whether the datasets were created from other prediction algorithms, or are 'proven' phase separating proteins. Also, it was unclear how the authors distinguished between ID-PSPs and noID-PSPs for these datasets.
- In figure 5, immunofluorescence and in vitro condensation are used as proof of phase separation for 8 chosen predicted proteins.
 - o How were these 8 proteins chosen for further investigation? Please describe in the text.
 - o Negative and positive controls, as well as some quantification of the immunofluorescence data are necessary for this to be believable.
 - ♣ Negative control: Co-staining of EDC4 (primary and secondary antibody 568) with only secondary antibody of 488 (no primary) to demonstrate that the weak co-staining seen in PGM1 and VPS26B are not off-targets from the secondary.
 - ♣ Positive control: Staining of one or more proteins in the ID-PSP and noID-PSP datasets
 - ♣ Quantification: The shown example images are sufficient quality, but the argument that these proteins do phase separate would be more convincing with appropriate quantification. For example, the % of cells with EDC4-colocalizing puncta across 3 biological samples of the same cell type, or across multiple cell types.
 - o In the in vitro purified protein experiments, the amount of PEG used in the condensation buffer is quite high (15%). Standard in the field is more like 2%. I consider it essential to test either a negative control non-phase separating protein in the 15% PEG case, to show that this high amount of PEG does not induce phase separation of a non-PSP, and/or redoing the PGM1, SerpinB4, Rab31 and S100A7 with lower PEG in the buffer (e.g. 2%).
- Related to figure 5, I would appreciate a paragraph describing the difference between drivers and

passengers in phase separation, and single-component vs multi-component structures. Visualizing puncta is one piece of evidence that these proteins participate in phase separation, but does not distinguish between drivers and passengers (scaffolds/clients). In vitro purified protein condensation assays are stronger evidence of drivers of phase separation. Does PSPire have any indication of which proteins are more likely to be 'drivers'? Perhaps the strongest hits?
- Would also appreciate an additional discussion paragraph about the other future uses of SSUP feature predictions, other future uses of PSPire

Minor text comments:

Line 35: Phrasing "MLOs such as nucleoli and stress granules can concentrate proteins and nucleic acids at specific cellular sites without physical boundaries"
Change to 'without bounding membranes'

Line 36: Strange to mention brangwynnes name specifically in text, rewrite this sentence?

Line 37: Liquid -> liquid-like droplets

Line 47 'participate into' -> can form

Line 55, what are the parameters of 'IDR' here? To categorize yes vs no ID

Line 88 dataset -> datasets ?

Fig 5: Immunostaining—were these antibodies already available?

Did this study validate their specificity in any way?

Please provide more details in methods about the antibodies and conditions for immunofluorescence:

- I can't find info about abs117064, absin rabbit anti-PGM1 or abs117670 anti-TXNL4B on the company websites

- Are all these proteins expressed at relevant levels in the cell lines tested?

Comments on figures:

Figure 1. Please define acronyms in the figure legend (specifically SaPS and PdPS)

- The label 'IDR percentage' is confusing because the actual metric is length of region (in aa) with IDR score cutoff of 0.5.

- IDR_50 and IDR_60 are supposed to be percentages but have values below 0 and above 1?

- Please include statistical information in figure legend: what test was performed? What is n in each group?

Figure 2. in b, consider instead labeling in plain language in figure and referencing the name of the group in text. a/c/d are nice

Figure 3. is beautiful.

Fig 5. Similar comment as above: The images themselves are high enough quality but immunofluorescence needs some sort of quantification, as well as negative controls. E.g. PGM1 and VPS26B puncta are less obvious than TXNL4B and Serpin B4 puncta. Were these experiments repeated in multiple biological samples? Perhaps a metric like % of cells with >0.6 PCC across multiple biological samples would be more convincing. What do the 2 test proteins that were considered not phase separating look like?

Reviewer #2:

Remarks to the Author:

In the manuscript by Hou et al., the authors elucidate the development of PSPire, a sophisticated machine learning predictor specifically devised to synergistically incorporate both residue-level and structural-level attributes to enhance the precision in predicting PSPs. Distinctly deviating from

contemporary predictors such as PSAP and PhaSePred which predominantly depend on amino acid features, PSPire seamlessly integrates three-dimensional structural data, consequently achieving marked improvement in discerning noID-PSPs. This improvement underscores a salient relationship between non-IDR features founded on structured superficial regions and the inherent multivalency that characterizes the phase separation (PS) mechanism in structural domain-driven proteins. Notably, PSPire's predictive prowess was corroborated through meticulous biological experimental validation. Collectively, PSPire manifests considerable promise not only in the rigorous prediction of PS but also in advancing the identification of potential therapeutic targets associated with PS. For an enriched robustness of the study, I recommend that the authors address the subsequent concerns:

1. Upon perusal, it became apparent that the authors, in their quest to assess the efficacy of integrating structural data into PSP prediction, predominantly amalgamated top-importance features from leading predictors such as PSAP and PhaSePred into PSPire. When considering the biological validation outcomes, only 6 out of the 8 selected candidates were verified to form condensates. Would the inclusion of additional PS-correlated features augment the predictive accuracy of the PSPire model?
2. As delineated in the Methods section, the positive training dataset encompassed 259 PSPs, of which 195 were ID-PSPs and 64 were noID-PSPs. The positive testing dataset comprised 258 PSPs, with 194 being ID-PSPs and 64 being noID-PSPs. Given this disparity, could there be potential ramifications on the model's accuracy concerning the prediction of PS in non-ID proteins? If such an imbalance poses a detriment, refinements to the model are advised.
3. The authors employed SSUP to ascertain protein structures in the testing dataset, utilizing AlphaFold-predicted 3D structures. Considering the well-documented knowledge that AlphaFold-generated structures don't guarantee absolute accuracy, has the potential margin of error been factored into the PSPire model?
4. Within the biological validation segment, the authors identified 8 PS candidates, of which 6 were empirically validated to form condensates both in cells and in vitro. To conclusively satisfy PS criteria, I propose the presentation of FRAP data pertaining to PGM1, SerpinB4, Rab31, and S100A7.
5. There exists a plethora of literature (e.g., [10.1021/acs.biochem.1c00376](https://doi.org/10.1021/acs.biochem.1c00376), [10.1007/s11427-020-1702-x](https://doi.org/10.1007/s11427-020-1702-x)) that offers profound insights into protein interactions within biomolecular condensates, an aspect pivotal to this discipline.

Reviewer #3:

Remarks to the Author:

This article by Hou et al addresses the development of a machine learning tool to predict phase separating globular proteins that do not contain intrinsically disordered regions. The tool has two main applications: 1) it considers residue and structure level features and 2) it is the first tool which predicts phase separating globular proteins without IDRs. The authors initially focused on the discrimination of No-ID-PSPs from ID-PSPs and No-PSPs. The authors then focused on multivalency and transient interactions responsible for PS formation derived from the established spacers and stickers model for phase separation of proteins. Next residue and structure level features were extracted to predict noID-PSPs that phase separate. PSPire predicted 8 proteins to phase separate, experimental validation confirmed that 6 were able to phase separate in vitro and in cells. Overall, the method is interesting but it is difficult to follow how different steps in the pipeline were determined to be important or not (see comments below) making it a challenge to evaluate how this method works to predict noID-PSPs and what signal is used to distinguish the features that identify bona fide noID-PSP. The experimental validation of the hits is very weak – several of the hits are membrane associated proteins that may be able to cluster for other reasons. Given all the gaps in the manuscript, I do not recommend this manuscript for publication in Nat Comms.

Major Comments

1. The full length protein feature comparison of known predictors and known datasets identifies Leu frequency that can distinguish noID-PSPs from ID-PSPs and No-PSPs but looking at the data IDR50 and IDR60 can also have some discriminatory power. Its not clear why Leucine frequency is important, a physico chemical rational for this must be provided. Also, why inclusion of Ile+Leu doesn't discriminate. Are the frequencies of these residues different in IDR vs globular proteins?
2. The authors identified SSUP residues using the alpha-fold generated 3D model, for this the authors used pLDDT score for separating IDR region, then DSSP used to identify secondary structure and finally based on relative surface area categorized SSUP residue is exposed and buried (25%). There is no data in the manuscript to show this – relative surface exposure data must be provided (related to Figure 2e). Additionally, model information based on the pLDDT criteria must be provided.
3. There is no explanation of how the Alpha-fold data was analyzed. What dataset was used? What were the pLDDT scores of corresponding models? Why did the authors choose a cutoff 50 for the pLDDT score. There is no DSSP data for the AF models used in the manuscript. This needs to be explained and the data has to be provided with the manuscript.
4. SSUP residues are featurized considered as IDR independent features. The SSUP residues utilized in describing stickers and spacers model are focused towards charged residues in sticker interaction in PS and based on 14 Ang interaction cutoff cluster them together, identified stickers and compared in No-ID-PSPs , ID-PSPs and No-PSPs. What were the models and tools used here?
5. Phosphorylation and secondary structure (helix) frequency is used to distinguish ID-PSPs and No-ID-PSPs using default parameters in XGBoost. Were the parameters optimized? How did the authors deal with overfitting the data with XGBoost?
6. Out of 8 identified hits, 2 of these proteins were ID-PSPs (Rab31 and VPS26B) but were included in the nonIDPSP set. So only 4 hits were validated as noID-PSPs. 50% accuracy.
7. How is the PS score calculated? This should be defined.
8. How did the authors select the 6 candidate hits from all Non-ID PSP? The selection criteria for hits need to be explicitly stated in the results and methods. For example, the top hit based on PS score (ANXA3) failed in the experiment.
9. How did the authors analyze the data to produce the plots in Figure 2b for the feature comparison. The authors need to provide a lot more detail for how this analysis was carried out. Also, the authors need to provide model information, sensitivity, specificity and a table that includes the accuracy comparison with the other models.
10. The authors claim to identify noIDPSPs based on SSUP features (electrostatics and helicity) but in the text (lines 139-143) describe how the disordered N-terminus interacts with the coiled-coiled domain via electrostatic interactions.
11. Why did the authors not consider hydrophobic interactions and only considered electrostatic interaction for the sticker-related feature?
12. It is not clear why IDR features (charged IDR and cys IDR) are compared to SSUP feature – one is linked to a IDR and the other to an globular protein (supp Figure 3). The selected feature which distinguished the No-ID-PSPs , ID-PSPs should be null in one compare to other. The efficiency of PSPire is dependent on SSUP prediction which is based on exposure of residues and it's implications on structure superficial region.
13. Experimental validation is quite weak and needs to be improved to provide convincing data that the identified hits behave according to the predicted features. For the cell experiments, its not clear why the authors would expect that the noIDPSPs would colocalize with stress granules (SG) (G3BP1) or P-bodies (PB) (EDC4)? The prediction is based on intrinsic properties of the protein and stress granules and P-bodies are complex assemblies composed of hundreds of proteins (and other molecules that include RNA). The fact that these proteins co-localize with these SG and PB suggests that something more complex is going on. Additionally, Rab31 and VPS26B are proteins linked to vesicular trafficking and would likely cluster on membranes on their own – again not clear why they would colocalize with SGs/PBs.
14. For the in vitro validation, given the prediction of specific features important for the property of the given protein a direct test of the prediction is important. For example, if the interaction is predicted to be electrostatic a phase diagram using ionic strength as one variable is essential. An alternative would be to test mutants at key feature sites and show loss of ability to phase separate. Again, the goal of the validation is not necessarily whether the protein assembles but rather that the predicted features used to identify them as a hit are important for the property.
15. The authors should also test proteins that are predicted to no phase separate – negative

controls are really important and will provide further confirmation that the model picks up features that drive phase separation properties.

Minor comment

1. In the first analysis to discriminate noID-PSPs from ID-PSPs and No-PSPs the supportive data provided is missing key information to interpret the conclusions. For example, it is missing False positive and False negative scores. The authors should also provide the table that includes the details of tools for each machine learning algorithm.
2. Structured superficial region (SSUP) of PSPs : significant role in multivalency involved in phase separation. SSUP excludes IDRs, features from SSUP are IDR independent.
3. Please include a figure that describes the sticker and spacer model for different types of protein-protein interactions that dictate phase separation. This helps explain what the overall goal of the manuscript is.
4. What SSUP features differentiate ID-PSPs and noIDPSPs?
5. It would be really useful to include a diagram that illustrates the overall method and approach.
6. In Figure 4c, PdPs from PhasePred is yielding a more significant score for a few datasets in comparison to PSPire. This needs to be explained why PhasePred (PdPs) is better than PSPire in the MLO dataset.
7. There needs to be a clear explanation/description for why no-IDPSP protein also contain the ID region but the important feature for PS formation is its secondary structure/domain rather than ID region. This is really confusing.
8. Alpha-fold predictions are notoriously bad for IDR and sometimes can be annotated as secondary structure. This of course will change how PSPire uses secondary structure inferences if the model is incorrectly annotated.

Reviewer #1:

This manuscript by Hou & Hu et al., sets out to create a better computational predictor of phase separating proteins, particularly for those that do not contain significant disordered content. In the field, emphasis has been placed on intrinsically disordered domains for driving biomolecular condensation in living cells, which has translated into emphasis in the features utilized for training predictors of potential phase separating proteins. However, many examples of phase separating proteins exist that do not require or utilize disordered regions for phase separation. In this study, Hou & Hu et al. created an alternate metric for predicting self-interaction propensity (“stickers” in the stickers-and-spacers model) that works for well-folded proteins in addition to disordered proteins. They employ alpha fold to identify folded and disordered regions, which can be modified and updated as alpha fold itself progresses. They then choose 8 candidate proteins predicted to have high phase separation potential from their prediction algorithm, and find that 6 of these 8 localize to punctate structures in living cells and demix from aqueous solution with a PEG crowder in vitro, so they conclude this supports their capability of phase separation.

The premise of this manuscript is broadly interesting for audiences in the fields of computational biology, biomolecular condensation and protein structure prediction. I commend the authors for building the PSPire tool, which represents an advance in our ability to comprehensively predict phase separating proteins, and will be useful for hypothesis generation in future studies.

The weaknesses of the manuscript lie in communicating the details of their training and testing data sets, and in demonstrating that the predicted phase-separating proteins indeed are capable of condensation. See below for suggestions on how to strengthen these points. Overall, I recommend publication with minor edits.

Major comments:

1. Please include a sentence or two in the main text describing how the authors created the training and testing datasets; in this draft these important parameters are only written in the methods. Specifically, it was unclear whether the datasets were created from other prediction algorithms, or are ‘proven’ phase separating proteins. Also, it was unclear how the authors distinguished between ID-PSPs and noID-PSPs for these datasets.

Response: Thanks for the reviewer’s constructive suggestions. We added a description of the dataset collection process at the beginning of the updated Results section. The training and testing datasets were not created by other prediction algorithms. All phase-separating proteins (PSPs) in the training and testing datasets were validated by *in vivo* or *in vitro* experiments or by the identification of membraneless compartments.

PSPs were classified into ID-PSPs and noID-PSPs based on the presence or absence of intrinsically disordered regions (IDRs). The criteria for determining IDRs are detailed in the Methods section. We added a description of the method of distinguishing between ID-PSPs and noID-PSPs in the Methods section.

2. In figure 5, immunofluorescence and in vitro condensation are used as proof of phase separation for 8 chosen predicted proteins.

- *How were these 8 proteins chosen for further investigation? Please describe in the text.*

Response: We chose these proteins for experimental validation based on the following criteria: 1) predicted PS score, 2) protein size, 3) literature support of association with the biological processes of interest, 4) the availability of antibodies for immunofluorescence, 5) the feasibility of expressing and purifying proteins for in vitro condensation assays. We added the details of the selection criteria in the updated Results section.

- *Negative and positive controls, as well as some quantification of the immunofluorescence data are necessary for this to be believable.*
 - *Negative control: Co-staining of EDC4 (primary and secondary antibody 568) with only secondary antibody of 488 (no primary) to demonstrate that the weak co-staining seen in PGM1 and VPS26B are not off-targets from the secondary.*
 - *Positive control: Staining of one or more proteins in the ID-PSP and noID-PSP datasets.*

Response: Thanks for the insightful comments provided by the reviewer. According to the reviewer's suggestions, we performed a series of staining procedures on HeLa cells.

In this study, we co-stained EDC4 (the mark of P-body) and candidate PSPs using their respective primary antibodies, followed by secondary antibodies (Alexa Fluor 568 for EDC4 and Alexa Fluor 488 for candidate PSPs) (Fig. 5c). For the issue of negative control, we included controls where only the secondary antibody Alexa Fluor 488 was used, in order to evaluate the specificity of the Alexa Fluor 488. As shown in Fig. R1a, b, negligible fluorescence was observed in the 488 channels in the absence of the primary antibody. This finding clearly suggests minimal to no off-target binding by the secondary antibodies.

For the issue of positive control, we included two known PSPs, TARDBP and TBK1. TARDBP is a known ID-PSP. Post arsenite treatment, cells were immunostained for TARDBP and G3BP1 (the mark of stress granules). As shown in Fig. R1c, the results indicated a co-localization of TARDBP with G3BP1, aligning with previous reports. TBK1 is a known noID-PSP. We examined the co-localization of TBK1 and LC3 (the mark of autophagosome) in cells treated with bafilomycin. Fig. R1d demonstrated that TBK1 formed distinct puncta and co-localizes with LC3 in the cytoplasm. This finding strongly supports the quality of our immunofluorescence data.

Figure R1. Immunofluorescence of negative and positive controls in HeLa cells. **a and b**, Confocal microscopy image of HeLa cells post 500 μ M sodium arsenite treatment for 1 hour, followed by incubation with Alexa Fluor 488-conjugated secondary antibody (green) without primary antibody. **(a)** G3BP1 (red) and **(b)** EDC4 (red) were incubated with primary antibody followed by secondary antibody. G3BP1 and EDC4 mark the stress granules and the P-body, respectively. **c**, HeLa cells were exposed to 500 μ M sodium arsenite for 1 hour and immunostained for endogenous TARDBP (green) and G3BP1 (red). G3BP1 marks the stress granules. **d**, HeLa cells were treated with 50 nM bafilomycin for 10 hours and immunostained for endogenous TBK1 (green) and LC3 (red). LC3 marks the autophagosome. The nucleus was stained with 4',6-diamidino-2-phenylindole (DAPI; blue). Scale bars represent 5 μ m.

- *Quantification: The shown example images are sufficient quality, but the argument that these proteins do phase separate would be more convincing with appropriate quantification. For example, the % of cells with EDC4- colocalizing puncta across 3 biological samples of the same cell type, or across multiple cell types.*

Response: We acknowledge the reviewer's comment on the importance of quantitative analysis. In concurrence with the suggestion that quantifying cells with co-localizing puncta of EDC4 (the mark of P-body) or G3BP1 (the mark of stress granules) could augment the evidence for phase separation, we conducted a rigorous and systematic quantitative analysis.

To ensure the robustness of our data, we preformed three biological replicates for each protein of interest, namely SERPINB4, PGM1, TXNL4B, VPS26B, RAB31, and S100A7. We acquired approximately ten microscopic fields per sample to facilitate an unbiased and representative quantification. Our analysis revealed a substantial degree of co-localization for the following pairs: SERPINB4 and EDC4, PGM1 and EDC4, TXNL4B and EDC4, RAB31 and G3BP1, S100A7 and G3BP1 (Fig. R2a,b). Conversely, the immunofluorescence assay for VPS26B demonstrated a relatively modest co-localization with EDC4 (Fig. R2a). The quantitative data affirmed a significant association between our experimentally validated protein candidates and canonical membraneless organelles.

The quantification analysis of the co-localization percentages was also added in the revised manuscript and was presented as the updated Supplementary Fig. 6a.

Figure R2. Quantification analysis of the co-localization between candidate PSPs and canonical membraneless organelles in HeLa cells. **a**, left, confocal images of endogenous SERPINB4, PGM1, TXNL4B, and VPS26B in HeLa cells. EDC4 (red) marks the P-body. Right, quantification of the co-localization percentages of endogenous SERPINB4, PGM1, TXNL4B, and VPS26B with EDC4 signal. $n = 30$ for co-localization percentage analysis. **b**, left, representative images of endogenous RAB31 and S100A7 in HeLa cells under stress conditions induced by sodium arsenite. G3BP1 (red) marks the stress granules. Right, quantification of the co-localization percentages of RAB31 and S100A7 with G3BP1 signal. Scale bars represent 5 μm .

- *In the in vitro purified protein experiments, the amount of PEG used in the condensation buffer is quite high (15%). Standard in the field is more like 2%. I consider it essential to test either a negative control non-phase separating protein in the 15% PEG case, to show that this high amount of PEG does not induce phase separation of a non-PSP, and/or redoing the PGM1, SerpinB4, Rab31 and S100A7 with lower PEG in the buffer (e.g. 2%).*

Response: Thanks for the reviewer's insightful comments concerning the PEG concentration employed in our protein phase separation assays. To address the concern raised, we revisited our assays with modified PEG concentrations. Specifically, we conducted additional experiments with SERPINB4, PGM1, RAB31, and S100A7 at 10% and 5% PEG. As shown in Fig. R3, the results demonstrated discernible phase separation at 10% PEG, evidenced by the formation of spherical droplets. At 5% PEG, however, the propensity for droplet formation was reduced, indicating a diminished phase separation capability.

We acknowledge that while a 2% PEG 20000 concentration is often standard¹, optimal conditions indeed require empirical determination for each protein system under investigation. For example, 10% PEG 8000 was used for the PS of TIA1², lactoferrin³, β -lactoglobulin³, lysozyme³, RNase A³, and Tau³. Additionally, 15% PEG 8000 was used for the PS of alpha-synuclein⁴, TIP60⁵, and SOD1⁶. We discussed the PEG concentration used in the *in vitro* purified protein experiments in the updated Methods section.

Figure R3. Candidate PSPs undergo phase separation *in vitro*. Representative confocal images of SERPINB4, S100A7, PGM1, and RAB31 following incubation with 15%, 10%, and 5% PEG 8000 in 20 mM Tris-HCl pH 7.5, 150 mM NaCl. Scale bars represent 5 μ m.

3. Related to figure 5, I would appreciate a paragraph describing the difference between drivers and passengers in phase separation, and single-component vs multi-component structures. Visualizing puncta is one piece of evidence that these proteins participate in phase separation, but does not distinguish between drivers and passengers (scaffolds/clients). In vitro purified protein condensation assays are stronger evidence of drivers of phase separation. Does PSPire have any indication of which proteins are more likely to be 'drivers'? Perhaps the strongest hits?

Response: Thanks for the reviewer's valuable suggestion. PSPire is not designed to indicate which proteins are more likely to be drivers or passengers of phase separation, mainly due to the composition of the training dataset. Among the four representative LLPS databases, only the DrLLPS database distinctly categorizes proteins as scaffolds, regulators, or clients. Besides, the PhaSePro database exclusively collected experimentally validated driver proteins. The combined total of 86 human scaffold proteins from the DrLLPS database and 55 driver proteins from the PhaSePro database amounts to 103 PSPs. Out of these 103 proteins, 102 are incorporated into the positive dataset for the final training of PSPire, making up approximately 20% of the total positive dataset. As a result, PSPire can capture the features of both drivers and passengers in phase separation. For example, 11 out of the 12 documented scaffold mouse proteins in DrLLPS were identified as candidate PSPs by PSPire and they were evenly distributed among the top hits. We acknowledge that *in vitro* purified protein condensation assays could indicate the potential role of these proteins as drivers within cells. We added a paragraph for this issue in the updated Discussion section.

4. Would also appreciate an additional discussion paragraph about the other future uses of SSUP feature predictions, other future uses of PSPire.

Response: Thanks for the reviewer's insightful suggestion. According to the reviewer's comment, we added a paragraph for this issue in the updated Discussion section as follows:

“SSUP residues, particularly those constituting stickers, offer sites where mutations might impact PS behavior, which could be valuable for further experimental validation and has the potential to aid in the identification of drug targets related to PS. Besides biological experiments, critical residues in SSUP can be further explored using molecular dynamics to uncover potential mechanisms driving PS. Leveraging these important features, PSPire reported the residue positions of SSUP and identified stickers as output.”

5. Minor text comments:

- 1) Line 35: Phrasing “MLOs such as nucleoli and stress granules can concentrate proteins and nucleic acids at specific cellular sites without physical boundaries”. Change to ‘without bounding membranes’
- 2) Line 36: Strange to mention brangwynnes name specifically in text, rewrite this sentence?
- 3) Line 37: Liquid -> liquid-like droplets
- 4) Line 47 ‘participate into’ -> can form
- 5) Line 55, what are the parameters of ‘IDR’ here? To categorize yes vs no ID
- 6) Line 88 dataset -> datasets?

Response: According to the reviewer's comments, we modified the texts in the updated manuscript.

6. Fig 5: Immunostaining—were these antibodies already available? Did this study validate their specificity in any way? Please provide more details in methods about the antibodies and conditions for immunofluorescence:

- I can't find info about abs117064, absin rabbit anti-PGM1 or abs117670 anti-TXNL4B on the company websites
- Are all these proteins expressed at relevant levels in the cell lines tested?

Response: Since the specificity of the antibodies is not provided by the sourced company, we validated it through Western blot analysis (Fig. R4a). We analyzed the whole cell lysates of HeLa cells and showed the whole PVDF membrane for each protein. There was only one dominant band for each of ANXA3, S100A7, PGM1, and TXNL4B, indicating the distinguished specificity of the corresponding antibodies. For SERPINB4 and VPS26B, the bands of interest were hardly visible, suggesting the low specificity of their antibodies. For RAB31, there was another band in the membrane, indicating the non-specific binding. Overall, the specificity of the antibodies used in this study is generally reasonable.

The websites of the anti-PGM1 and anti-TXNL4B are as follows:

<https://www.absin.cn/rabbit-pgm1-polyclonal-antibody/abs117064.html>

<https://www.absin.cn/rabbit-txnl4b-polyclonal-antibody/abs117670.html>

For the quantitative assessment of protein expression levels for these proteins in the tested cell line, we utilized median protein expression values (MS1-iBAQ) obtained from the ProteomicsDB database⁷⁻⁹, a comprehensive and integrative database for proteomics research. The result showed that the expression levels of these proteins are largely comparable (Fig. R4b).

Figure R4. a, Western blot of endogenous ANXA3, RAB31, S100A7, PGM1, TXNL4B, SERPINB4, and VPS26B in HeLa cells. The proteins of interest are denoted by bands within the red dashed boxes. **b**, Median protein expression values (MS1-iBAQ) for ANXA3, RAB31, S100A7, PGM1, TXNL4B, SERPINB4, and VPS26B in HeLa cells, data from ProteomicsDB database.

7. Comments on figures:

- * *Figure 1. Please define acronyms in the figure legend (specifically SaPS and PdPS)*
 - *The label 'IDR percentage' is confusing because the actual metric is length of region (in aa) with IDR score cutoff of 0.5.*
 - *IDR_50 and IDR_60 are supposed to be percentages but have values below 0 and above 1?*
 - *Please include statistical information in figure legend: what test was performed? What is n in each group?*

Response: According to the reviewer's comments, we modified the text as follows:

- The PhaSePred tool includes two models: SaPS for self-assembling proteins and PdPS for partner-dependent proteins. We included the explanations for the acronyms in the legend of Fig. 1, Fig. 4c, and Supplementary Fig. 1a.
- We changed the previous 'IDR percentage' to 'IDR length'.
- The values of IDR_50 and IDR_60 range between 0 and 1. For Fig. 1b, we utilized the 'sns.violinplot' function from the Seaborn library, which offers a smoothed representation of the data distribution through kernel density estimation (KDE). Due to the nature of this smoothing, the estimated distribution can occasionally appear to extend beyond the actual range of the data.
- P-values were calculated using the two-sided Mann-Whitney U test and the comparison was conducted on the union of training and testing datasets which contained 389 ID-PSPs, 128 noID-PSPs, and 10,284 non-PSPs. We added the statistical details to the figure legend and Methods section.

** Figure 2. in b, consider instead labeling in plain language in figure and referencing the name of the group in text. a/c/d are nice*

Response: According to the reviewer's comment, we relabeled the group names in Fig. 1b by including amino acid information for clearer representation. Similarly, we also updated Fig. 2b, and Supplementary Fig. 1b in the revised manuscript.

** Figure 3. is beautiful.*

Response: We appreciate the reviewer for this feedback.

** Fig 5. Similar comment as above: The images themselves are high enough quality but immunofluorescence needs some sort of quantification, as well as negative controls. E.g. PGM1 and VPS26B puncta are less obvious than TXNL4B and Serpin B4 puncta. Were these experiments repeated in multiple biological samples? Perhaps a metric like % of cells with >0.6 PCC across multiple biological samples would be more convincing. What do the 2 test proteins that were considered not phase separating look like?*

Response: As mentioned above, we performed three biological replicates for each protein and quantified their co-localization percentage with G3BP1 or EDC4 (Fig. R2). The quantification analysis of the co-localization percentages was added in the revised manuscript and was presented as the updated Supplementary Fig. 6a.

We also included the immunofluorescence images of the two proteins considered not to phase separate (Fig. R5), which demonstrated that CKMT2 and ANXA3 display a diffuse cytoplasmic distribution without the formation of discrete puncta. Fig. R5 was presented as the updated Supplementary Fig. 6c, d.

Figure R5 (updated Supplementary Figure 6c and d). Endogenous CKMT and ANXA3 cannot formed puncta with or without treatment. a, Confocal microscopy images of CKMT2 in HeLa cells following exposure to sodium arsenite stress (top panel) and in untreated cells (bottom panel). **b,** Confocal microscopy images of ANXA3 in HeLa cells subjected to sodium arsenite-induced stress (top panel) and in control conditions (bottom panel). In both panels, scale bars correspond to 5 μ m.

Reviewer #2:

In the manuscript by Hou et al., the authors elucidate the development of PSPire, a sophisticated machine learning predictor specifically devised to synergistically incorporate both residue-level and structural-level attributes to enhance the precision in predicting PSPs. Distinctly deviating from contemporary predictors such as PSAP and PhaSePred which predominantly depend on amino acid features, PSPire seamlessly integrates three-dimensional structural data, consequently achieving marked improvement in discerning noID-PSPs. This improvement underscores a salient relationship between non-IDR features founded on structured superficial regions and the inherent multivalency that characterizes the phase separation (PS) mechanism in structural domain-driven proteins. Notably, PSPire's predictive prowess was corroborated through meticulous biological experimental validation. Collectively, PSPire manifests considerable promise not only in the rigorous prediction of PS but also in advancing the identification of potential therapeutic targets associated with PS. For an enriched robustness of the study, I recommend that the authors address the subsequent concerns:

1. Upon perusal, it became apparent that the authors, in their quest to assess the efficacy of integrating structural data into PSP prediction, predominantly amalgamated top-importance features from leading predictors such as PSAP and PhaSePred into PSPire. When considering the biological validation outcomes, only 6 out of the 8 selected candidates were verified to form condensates. Would the inclusion of additional PS-correlated features augment the predictive accuracy of the PSPire model?

Response: We appreciate the reviewer's constructive feedback. To investigate the effectiveness of additional PS-related features, we included all features from PSAP that can be calculated on both structured superficial regions (SSUP) and intrinsically disordered regions (IDRs). In addition, we calculated the average polarity values for residues within IDRs and SSUP separately. Incorporating the above features led to enhanced PSPire performance across all datasets, as shown in Table R1. According to the reviewer's comment, we updated PSPire by incorporating these PS-related features in the revision. We also updated Fig. 4 in the revised manuscript.

Table R1. Performance comparison between original and updated versions of PSPire.

PSPs	Dataset	Type	PSPire (original)	PSPire (updated)	
noID-PSPs	Testing dataset	ROC	0.84	0.84	
		PRC	0.19	0.24	
	G3BP1 proximity labelling	ROC	0.90	0.93	
		PRC	0.52	0.66	
	RNAgranuleDB	ROC	0.89	0.90	
			0.20	0.28	
		PhaSepDB_MLO	ROC	0.80	0.80
			PRC	0.68	0.71
	DrLLPS_MLO	ROC	0.84	0.85	
		PRC	0.70	0.74	
ID-PSPs	Testing dataset	ROC	0.85	0.86	
		PRC	0.47	0.51	
	G3BP1 proximity labelling	ROC	0.89	0.91	
		PRC	0.49	0.58	

RNAgranuleDB	ROC	0.82	0.84
	PRC	0.44	0.48
PhaSepDB_MLO	ROC	0.69	0.72
	PRC	0.77	0.79
DrLLPS_MLO	ROC	0.73	0.75
	PRC	0.75	0.78

2. As delineated in the Methods section, the positive training dataset encompassed 259 PSPs, of which 195 were ID- PSPs and 64 were noID-PSPs. The positive testing dataset comprised 258 PSPs, with 194 being ID-PSPs and 64 being noID-PSPs. Given this disparity, could there be potential ramifications on the model's accuracy concerning the prediction of PS in non-ID proteins? If such an imbalance poses a detriment, refinements to the model are advised.

Response: We agree with the reviewer that sample imbalance influences the performances of machine learning models. In this study, to reduce the effect of sample imbalance, we assigned sample weights to ID-PSPs, noID-PSPs, and non-PSPs in the training dataset based on each type's inverse frequency at the XGBoost model fitting process. We described the details in the Methods section.

3. The authors employed SSUP to ascertain protein structures in the testing dataset, utilizing AlphaFold-predicted 3D structures. Considering the well-documented knowledge that AlphaFold-generated structures don't guarantee absolute accuracy, has the potential margin of error been factored into the PSPire model?

Response: We appreciate the reviewer for raising this important issue. We discussed this issue in the Discussion section. Since the calculation of structured superficial regions relies on protein structures, the efficacy of PSPire is closely tied to the precision of the predicted structures. Hence, users are advised to interpret the predicted scores with discretion. Alternatively, users are encouraged to provide their own refined protein structures for enhanced accuracy in predictions.

4. Within the biological validation segment, the authors identified 8 PS candidates, of which 6 were empirically validated to form condensates both in cells and in vitro. To conclusively satisfy PS criteria, I propose the presentation of FRAP data pertaining to PGM1, SerpinB4, Rab31, and S100A7.

Response: We appreciate the reviewer's suggestion to enhance the validation by using FRAP analysis. According to the reviewer's suggestion, we performed FRAP experiments on PGM1, SERPINB4, RAB31, and S100A7 after labeling these proteins with fluorescent dyes (Fig. R6). The results demonstrated that PGM1, RAB31, and S100A7 exhibited a moderate fluorescence recovery rate within 240 seconds, while SERPINB4 displayed a lower recovery rate, suggesting relatively limited dynamics in the condensates of SERPINB upon PEG induction. The results supported the dynamic nature of the condensates formed by PSPire-predicted candidate proteins. We added the results as Supplementary Fig. 6f-i, and the details of FRAP experiments in the Methods section of the revised manuscript.

Figure R6 (updated Supplementary Figure 6f-i). The dynamic of PGM1, SERPINB4, RAB31, and S100A7 condensates. The top panel presents confocal FRAP series for (a) PGM1, (b) RAB31, (c) S100A7, and (d) SERPINB4 in a buffer containing 20 mM Tris-HCl (pH 7.5), 150 mM NaCl, and 10% PEG 8000. The bottom panel shows corresponding fluorescence recovery over time for (a) PGM1, (b) RAB31, (c) S100A7, and (d) SERPINB4, post-photobleaching. Data shown are means \pm SD, $n = 3$. Scale bars, 2 μm .

5. *There exists a plethora of literature (e.g., 10.1021/acs.biochem.1c00376, 10.1007/s11427-020-1702-x) that offers profound insights into protein interactions within biomolecular condensates, an aspect pivotal to this discipline.*

Response: Thanks for the reviewer's insightful suggestion. We discussed this issue in the updated Discussion section as follows:

“Multivalent interactions driving phase separation involve not only IDR-driven nonspecific interactions but also widely concern modular domain-mediated specific interactions^{10,11}. However, most existing PSP predictors displayed a marked bias towards proteins with high IDR contents, resulting in suboptimal performance when predicting noID-PSPs. To address this, we introduced non-IDR features based on SSUP to complement IDR-related features.”

Reviewer #3:

This article by Hou et al addresses the development of a machine learning tool to predict phase separating globular proteins that do not contain intrinsically disordered regions. The tool has two main applications: 1) it considers residue and structure level features and 2) it is the first tool which predicts phase separating globular proteins without IDRs. The authors initially focused on the discrimination of No-ID-PSPs from ID-PSPs and No-PSPs. The authors then focused on multivalency and transient interactions responsible for PS formation derived from the established spacers and stickers model for phase separation of proteins. Next residue and structure level features were extracted to predict noID-PSPs that phase separate. PSPire predicted 8 proteins to phase separate, experimental validation confirmed that 6 were able to phase separate in vitro and in cells. Overall, the method is interesting but it is difficult to follow how different steps in the pipeline were determined to be important or not (see comments below) making it a challenge to evaluate how this method works to predict noID-PSPs and what signal is used to distinguish the features that identify bona fide noID-PSP. The experimental validation of the hits is very weak – several of the hits are membrane associated proteins that may be able to cluster for other reasons. Given all the gaps in the manuscript, I do not recommend this manuscript for publication in Nat Comms.

Major Comments:

1. The full length protein feature comparison of known predictors and known datasets identifies Leu frequency that can distinguish noID-PSPs from ID-PSPs and No-PSPs but looking at the data IDR50 and IDR60 can also have some discriminatory power. Its not clear why Leucine frequency is important, a physico chemical rational for this must be provided.

Response: Mierlo *et al.* reported a depletion of leucine not only in human PSPs but also in known PSPs in different species¹². In our study, we separated the known PSPs into ID-PSPs and noID-PSPs, and observed the depletion of leucine in both types. It should be noticed that the fraction of leucine is a ratio within the entire protein sequence, and PSPs are depleted of leucine does not mean that leucine is not important in phase separation. Saar *et al.* reported that although PSPs are less hydrophobic, sufficient hydrophobic contents are required to reach the lowest saturation concentrations¹³.

** Also, why inclusion of Ile+Leu doesn't discriminate.*

Response: The fraction of Leucine can simultaneously distinguish ID-PSPs and noID-PSPs from non-PSPs. However, the fraction of isoleucine demonstrated opposing tendencies for ID-PSPs and noID-PSPs compared to non-PSPs. As a result, the “group_Xle” feature (*i.e.*, the fraction of Xle group, which includes Leu and Ile) did not show a statistically significant difference between noID-PSPs and non-PSPs, as shown in Fig. R7.

Figure R7. Comparison of fractions of Leu, Ile, and Xle group between the two types of PSPs (ID-PSPs and noID-PSPs) and non-PSPs. The three features calculated on the whole protein sequence are fraction_L (*i.e.*, the fraction of leucine), fraction_I (*i.e.*, the fraction of isoleucine), and group_Xle (*i.e.*, the fraction of Xle group which includes leucine and isoleucine).

* *Are the frequencies of these residues different in IDR vs globular proteins?*

Response: According to the reviewer's comment, we divided each ID-PSP into IDR fragments and non-IDR fragments, and compared the fractions of Leu, Ile, and Xle group among IDR, non-IDR fragments of ID-PSPs and noID-PSPs. As shown in Fig. R8, IDR fragments of ID-PSPs displayed significantly lower fractions of Leu, Ile, and Xle group.

Figure R8. Comparison of fractions of Leu, Ile, and Xle group among IDR, non-IDR fragments of ID-PSPs and noID-PSPs. The three features calculated on the selected fragment or whole protein sequence are fraction_L (*i.e.*, the fraction of leucine), fraction_I (*i.e.*, the fraction of isoleucine), and group_Xle (*i.e.*, the fraction of Xle group which includes leucine and isoleucine).

2. *The authors identified SSUP residues using the alpha-fold generated 3D model, for this the authors used pLDDT score for separating IDR region, then DSSP used to identify secondary structure and finally based on relative surface area categorized SSUP residue is exposed and buried (25%). There is no data in the manuscript to show this – relative surface exposure data must be provided (related to Figure 2e).*

Response: We appreciate the reviewer for this constructive comment. The relative surface exposure data of proteins in the human proteome can be downloaded from the GitHub repository (<https://github.com/TongjiZhanglab/PSPire>). We added the above link for downloading the relative surface exposure data in the revised manuscript.

* *Additionally, model information based on the pLDDT criteria must be provided.*

Response: We used a threshold of 50 for pLDDT scores to determine the intrinsically disordered regions (IDRs), *i.e.*, a residue with a score lower than 50 was considered disordered. We included the detailed information in the Methods section.

3. There is no explanation of how the Alpha-fold data was analyzed.

Response: The AlphaFold DB website provides PDB format files. We analyzed the downloaded PDB format files to determine intrinsically disordered regions (IDRs), structured superficial regions (SSUP), the secondary structure state of each residue, and sticker-related features.

To determine IDRs, we extracted pLDDT scores of each protein from PDB format files. To determine SSUP, we calculated the relative solvent accessible surface areas (RSA) based on the information from PDB format files by using the PSAIA program. To calculate the secondary structure state, we applied the DSSP program to PDB format files. To compute stickers, we extracted the C α 3D coordinates from PDB format files. We included the analysis process details in the Methods section.

** What dataset was used?*

Response: We downloaded the dataset (identifier: UP000005640) from the AlphaFold DB website, which contains PDB and mmCIF format files of 23,391 human proteins. The downloaded PDB format files were used in this study.

** What were the pLDDT scores of corresponding models?*

Response: The pLDDT score was introduced by the AlphaFold model, which reflected per-residue level confidence of the predicted protein structure. As Tunyasuvunakool *et al.* reported that a small pLDDT score is a strong predictor of disordered¹⁴, we utilized the score to determine IDRs in this study.

** Why did the authors choose a cutoff 50 for the pLDDT score.*

Response: Tunyasuvunakool *et al.* reported that pLDDT < 50 is a reasonably strong predictor of disordered¹⁴, which motivated us to utilize 50 as the threshold for the pLDDT scores.

** There is no DSSP data for the AF models used in the manuscript. This needs to be explained and the data has to be provided with the manuscript.*

Response: The secondary structure states were calculated by applying the DSSP program to PDB format files. The secondary structure state of each residue for proteins in the human proteome can be downloaded from the GitHub repository (<https://github.com/TongjiZhanglab/PSPire>). We added the above link for downloading the secondary structure states data in the revised manuscript.

4. SSUP residues are featurized considered as IDR independent features. The SSUP residues utilized in describing stickers and spacers model are focused towards charged residues in sticker interaction in PS and based on 14 Ang interaction cutoff cluster them together, identified stickers and compared in No-ID-PSPs, ID-PSPs and No-PSPs. What were the models and tools used here?

Response: To identify stickers, we applied the PyMOL software to calculate the net charge index of each SSUP residue by including proximate residues within a defined distance (14 Å used in this study). We then applied hierarchical clustering implemented in the Python scipy package to cluster the SSUP residues. We included the details in the Methods section.

5. Phosphorylation and secondary structure (helix) frequency is used to distinguish ID-PSPs and No-ID-PSPs using default parameters in XGBoost. Were the parameters optimized? How did the authors deal with overfitting the data with XGBoost?

Response: We appreciate the reviewer for raising this important issue. In our original manuscript, the XGBoost classifier trained using default parameters performed well on multiple independent datasets, suggesting that the model was likely not overfitting. Nevertheless, according to the reviewer's comment, we optimized hyperparameters through five-fold cross-validation to ensure that our model does not learn the noise in the training data and implemented the following strategies to prevent overfitting.

- We adjusted the 'max_depth' parameter to control the depth of the trees and 'min_child_weight' parameter to set the minimum sum of instance weight needed in a child, preventing the model from being overly complex.
- We optimized the 'learning_rate' parameter to ensure that each step in the gradient descent is small, allowing the model to learn gradually and avoid overfitting.
- We adjusted the 'n_estimators' parameter to control the number of trees in the model.
- We set the 'subsample' parameter to use a fraction of the data for each tree, and 'colsample_bytree' parameter to select a subset of features, both of which introduce randomness and prevent the model from fitting too closely to the training data.
- We incorporated the 'gamma' parameter, which specifies the minimum loss reduction required to make a further split on a leaf node of the tree. This helps in pruning and controlling the size of the trees, making the model more conservative.
- We implemented the regularization terms 'lambda' (L2 regularization) and 'alpha' (L1 regularization) to add penalties on the complexity of the model, smoothing the final learned weights to further prevent overfitting.

We implemented the strategies to prevent overfitting in the updated version of PSPire, and added the details in the updated Methods section.

6. Out of 8 identified hits, 2 of these proteins were ID-PSPs (Rab31 and VPS26B) but were included in the nonIDPSP set. So only 4 hits were validated as noID-PSPs. 50% accuracy.

Response: To strengthen the results of this study, during the revision, we validated three more noID-PSP candidates (HPRT1, H3C15, and ACTR2). We generated GFP-tagged constructs of these proteins and expressed them in HeLa cells. Immunostaining images exhibited distinct phase separation behaviors, with HPRT1 and ACTR2 forming cytoplasmic puncta, and H3C15 assembling into larger nuclear condensates, as shown in Fig. R9 (the updated Fig. 5a). We included the validation results in the updated Results section.

Figure R9 (updated Figure 5a). Validation of three PSP candidates. Confocal images of GFP-ACTR2, GFP-H3C15, and GFP-HPRT1 overexpressing in HeLa cells. Condensates were indicated by white arrows. Scale bars, 5 μm .

7. How is the PS score calculated? This should be defined.

Response: For a binary classification task, XGBoost can output a probability score for each instance in the dataset. The score represents the probability that a given instance belongs to the positive class (*i.e.*, PSPs in this study). During the training process, we performed ten rounds of training. The PS score of a protein was defined as the average probability score from the ten trained models. We included the detailed information in the Methods section.

8. How did the authors select the 6 candidate hits from all noID-PSPs? The selection criteria for hits need to be explicitly stated in the results and methods. For example, the top hit based on PS score (ANXA3) failed in the experiment.

Response: We chose these proteins for experimental validation based on the following criteria: 1) predicted PS score, 2) protein size, 3) literature support of association with the biological processes of interest, 4) the availability of antibodies for immunofluorescence, 5) the feasibility of expressing and purifying proteins for in vitro condensation assays. We added the details of the selection criteria in the updated Results section.

9. How did the authors analyze the data to produce the plots in Figure 2b for the feature comparison. The authors need to provide a lot more detail for how this analysis was carried out.

Response: We presented the data in Fig. 2b with the following procedures. First, for each protein in the training and testing dataset, we identified its SSUP residues by using the PSAIA program. Second, for each feature in Fig. 2b, SSUP residues were used to calculate the feature for each protein. Finally, for each feature, the violin plots were generated for ID-PSPs, noID-PSPs, and non-PSPs, respectively. The two-sided Mann-Whitney U test

was applied to calculate the statistical significance. We included the details in the Methods section.

** Also, the authors need to provide model information, sensitivity, specificity and a table that includes the accuracy comparison with the other models.*

Response: According to the reviewer’s comment, we added sensitivity, specificity, and accuracy of PSPire and other models, as shown in Table R2. Additionally, two other metrics were also calculated: the Matthews correlation coefficient (MCC) and the F1-score. The results were added to the updated Supplementary Table 2.

Table R2. Evaluation of PSPire and current predictors on six datasets.

PSPs	Dataset	Type	PSPire	SaPS	PdPS	PSPredictor	PSAP	FuzDrop	PSPer	PScore	catGRANULE	PLAAC
noID-PSPs	Testing dataset	MCC	0.22	0.08	0.10	-0.07	0.07	-0.11	-0.06	-0.06	0.02	-0.04
		F1-score	0.18	0.09	0.10	0.05	0.09	0.04	0.04	0.05	0.07	0.04
		Sensitivity	0.73	0.58	0.64	0.61	0.61	0.55	0.42	0.63	0.52	0.38
		Specificity	0.79	0.64	0.63	0.22	0.60	0.20	0.42	0.23	0.54	0.50
	Accuracy	0.79	0.64	0.63	0.24	0.60	0.21	0.42	0.24	0.54	0.49	
	G3BP1 proximity labelling	MCC	0.40	0.30	0.23	-0.08	0.14	-0.12	-0.10	-0.05	0.14	-0.08
		F1-score	0.36	0.28	0.22	0.08	0.16	0.07	0.06	0.08	0.15	0.07
		Sensitivity	0.90	0.80	0.78	0.77	0.65	0.56	0.42	0.68	0.72	0.53
		Specificity	0.83	0.79	0.71	0.11	0.65	0.22	0.37	0.22	0.59	0.30
	Accuracy	0.83	0.79	0.71	0.14	0.65	0.23	0.37	0.24	0.59	0.31	
	DACT1-particulate proteome	MCC	0.48	0.26	0.25	-0.12	0.17	-0.11	-0.03	-0.07	0.07	-0.13
		F1-score	0.48	0.26	0.25	0.05	0.19	0.09	0.10	0.09	0.14	0.07
		Sensitivity	0.84	0.76	0.80	0.23	0.72	0.59	0.48	0.59	0.62	0.40
		Specificity	0.89	0.74	0.70	0.51	0.62	0.22	0.46	0.28	0.53	0.35
	Accuracy	0.89	0.74	0.70	0.49	0.62	0.24	0.46	0.30	0.54	0.35	
	RNAgranuleDB	MCC	0.24	0.08	0.10	-0.06	0.08	-0.05	-0.04	-0.03	0.04	-0.04
		F1-score	0.17	0.09	0.09	0.03	0.07	0.04	0.04	0.04	0.06	0.04
		Sensitivity	0.86	0.51	0.61	0.33	0.71	0.61	0.45	0.71	0.59	0.35
		Specificity	0.78	0.73	0.68	0.47	0.55	0.24	0.43	0.22	0.53	0.51
	Accuracy	0.78	0.72	0.67	0.46	0.55	0.25	0.43	0.23	0.53	0.51	
	PhaSepDB_MLO	MCC	0.44	0.15	0.22	-0.27	0.14	-0.23	-0.11	-0.20	-0.01	-0.25
		F1-score	0.65	0.45	0.52	0.33	0.48	0.38	0.41	0.38	0.41	0.31
		Sensitivity	0.75	0.47	0.61	0.47	0.58	0.59	0.59	0.56	0.52	0.42
		Specificity	0.72	0.69	0.63	0.27	0.56	0.19	0.30	0.24	0.48	0.32
Accuracy	0.73	0.61	0.62	0.33	0.57	0.33	0.40	0.35	0.49	0.35		
DrLLPS_MLO	MCC	0.53	0.21	0.27	-0.29	0.22	-0.22	-0.12	-0.16	0.04	-0.19	
	F1-score	0.69	0.48	0.53	0.31	0.52	0.36	0.38	0.38	0.43	0.37	
	Sensitivity	0.77	0.54	0.66	0.48	0.70	0.60	0.59	0.65	0.61	0.60	
	Specificity	0.79	0.69	0.63	0.23	0.54	0.20	0.29	0.21	0.44	0.22	
Accuracy	0.78	0.64	0.64	0.31	0.59	0.32	0.39	0.35	0.49	0.34		
Testing dataset	MCC	0.40	0.34	0.38	0.20	0.25	0.18	0.11	0.21	0.19	0.20	
	F1-score	0.43	0.37	0.41	0.26	0.30	0.25	0.21	0.27	0.26	0.26	
	Sensitivity	0.77	0.77	0.75	0.68	0.75	0.66	0.56	0.65	0.69	0.66	
	Specificity	0.82	0.76	0.81	0.65	0.67	0.65	0.63	0.69	0.63	0.66	
Accuracy	0.82	0.76	0.81	0.65	0.68	0.65	0.63	0.69	0.64	0.66		
G3BP1 proximity labelling	MCC	0.43	0.35	0.30	0.12	0.26	0.05	0.08	0.05	0.20	0.07	
	F1-score	0.42	0.35	0.30	0.17	0.28	0.14	0.16	0.14	0.23	0.15	
	Sensitivity	0.86	0.80	0.83	0.71	0.69	0.62	0.56	0.60	0.69	0.47	
	Specificity	0.84	0.80	0.73	0.53	0.77	0.48	0.60	0.51	0.69	0.66	
Accuracy	0.84	0.80	0.74	0.55	0.76	0.48	0.60	0.51	0.69	0.65		
DACT1-particulate proteome	MCC	0.32	0.26	0.29	0.06	0.22	0.06	0.10	0.05	0.15	0.00	
	F1-score	0.29	0.24	0.26	0.11	0.21	0.11	0.13	0.10	0.15	0.08	
	Sensitivity	0.77	0.77	0.76	0.63	0.72	0.65	0.61	0.64	0.73	0.44	
	Specificity	0.83	0.77	0.81	0.53	0.75	0.50	0.62	0.48	0.61	0.56	
Accuracy	0.83	0.77	0.81	0.53	0.75	0.51	0.62	0.49	0.62	0.55		
RNAgranuleDB	MCC	0.37	0.36	0.35	0.24	0.29	0.20	0.18	0.24	0.29	0.25	
	F1-score	0.44	0.42	0.42	0.33	0.36	0.30	0.29	0.32	0.36	0.34	
	Sensitivity	0.73	0.77	0.70	0.67	0.74	0.72	0.65	0.75	0.76	0.68	
	Specificity	0.79	0.75	0.78	0.68	0.69	0.59	0.62	0.61	0.68	0.69	
Accuracy	0.78	0.75	0.77	0.68	0.69	0.61	0.62	0.63	0.69	0.69		
PhaSepDB_MLO	MCC	0.31	0.34	0.35	0.21	0.27	0.20	0.16	0.17	0.28	0.12	
	F1-score	0.67	0.67	0.71	0.66	0.69	0.67	0.61	0.65	0.69	0.60	
	Sensitivity	0.61	0.60	0.68	0.63	0.68	0.66	0.56	0.63	0.68	0.56	
	Specificity	0.71	0.74	0.67	0.58	0.59	0.55	0.60	0.54	0.60	0.56	
Accuracy	0.65	0.66	0.68	0.61	0.64	0.61	0.58	0.59	0.65	0.56		
DrLLPS_MLO	MCC	0.37	0.36	0.39	0.22	0.31	0.19	0.11	0.18	0.31	0.13	
	F1-score	0.69	0.68	0.72	0.64	0.67	0.63	0.57	0.62	0.69	0.59	
	Sensitivity	0.65	0.63	0.72	0.65	0.66	0.64	0.54	0.63	0.71	0.57	
	Specificity	0.72	0.73	0.67	0.56	0.65	0.54	0.57	0.54	0.60	0.56	
Accuracy	0.68	0.68	0.70	0.61	0.65	0.60	0.56	0.59	0.66	0.57		

10. The authors claim to identify noID-PSPs based on SSUP features (electrostatics and helicity) but in the text (lines 139-143) describe how the disordered N-terminus interacts with the coiled-coiled domain via electrostatic interactions.

Response: The example in Fig. 2d was used to illustrate the sticker calculation on SSUP, not limited to noID-PSPs. Phase separation of LINE-1 ORF1 was reported to be mediated through the stickers in the coiled-coil domain¹⁵. In Fig. 2d, we displayed the calculated stickers in the coiled-coil domain of LINE-1 ORF1, which is consistent with the previous report. We modified the text accordingly to avoid confusion.

11. Why did the authors not consider hydrophobic interactions and only considered electrostatic interaction for the sticker-related feature?

Response: We appreciate the reviewer for raising this comment. In our original manuscript, we focused on electrostatic interactions based on the following two points. First, the strength of electrostatic interactions, ranging from 2 to 15 kcal/mol, is typically greater than that of hydrophobic interactions, which range from 0.5 to 3 kcal/mol¹⁶. Second, the proportions of hydrophobic residues in SSUP were significantly lower in ID-PSPs and noID-PSPs than in non-PSPs, while the proportions of charged residues in SSUP were significantly higher in ID-PSPs and noID-PSPs compared to non-PSPs (as shown in Fig. 2b), suggesting that electrostatic interactions might be more prevalent than hydrophobic interactions for structural domain-driven phase separation.

During the revision, we attempted to modify the sticker identification method by incorporating hydrophobic residues. However, the incorporation of hydrophobic residues made no improvement in the prediction power of PSPire. Therefore, we did not include hydrophobic interactions for the sticker-related features. We discussed this issue in the updated Discussion section.

12. It is not clear why IDR features (charged IDR and cys IDR) are compared to SSUP feature – one is linked to a IDR and the other to an globular protein (supp Figure 3). The selected feature which distinguished the No-ID-PSPs, ID-PSPs should be null in one compare to other. The efficiency of PSPire is dependent on SSUP prediction which is based on exposure of residues and it's implications on structure superficial region.

Response: It should be clarified that the features shown in the original Supplementary Figure 3 (*i.e.*, the updated Supplementary Figure 4) were not meant to distinguish ID-PSPs and noID-PSPs. Instead, the top features of ID-PSPs were the important ones to distinguish ID-PSPs and non-PSPs, and the top ones included Phos frequency, IDR- and SSUP-related features. Similarly, the top features of noID-PSPs were the important ones in distinguishing noID-PSPs and non-PSPs, and the top ones included Phos frequency and SSUP-related features. We modified the text and figure legend to make it clear.

13. Experimental validation is quite weak and needs to be improved to provide convincing data that the identified hits behave according to the predicted features. For the cell experiments, its not clear why the authors would expect that the noIDPSPs would colocalize with stress granules (SG) (G3BP1) or P-bodies (PB) (EDC4)? The prediction is based on intrinsic properties of the protein and stress granules and P-bodies are complex assemblies composed of hundreds of proteins (and other molecules that include RNA). The

fact that these proteins co-localize with these SG and PB suggests that something more complex is going on. Additionally, Rab31 and VPS26B are proteins linked to vesicular trafficking and would likely cluster on membranes on their own – again not clear why they would colocalize with SGs/PBs.

Response: Thanks for the reviewer's insightful observations. We are grateful for the recognition of RAB31 and VPS26B as pivotal proteins in vesicular trafficking and their inherent tendency to cluster on membranes autonomously. We agree that their colocalization with SGs or PBs is not conclusively established in our research. This finding may indicate a fascinating facet of their function, warranting additional exploration, although it does not directly correlate with the primary emphasis of our current paper. In support of our findings, we reference external literature indicating that RAB31 can form condensates during FcγR-mediated phagocytosis¹⁷. This phenomenon not only aligns with our observations but also reinforces the hypothesis that RAB31 could undergo phase separation under certain cellular conditions.

We recognize the reviewer's caution in interpreting protein localization within condensates as definitive evidence of *in vivo* phase separation. It is indeed a valid consideration that RAB31 and VPS26B might be integrated into pre-formed condensates, rather than initiating spontaneous phase separation. To address this, we employed an *in vitro* assay with purified proteins, as illustrated in Fig. R10. The results from this assay demonstrate that RAB31 is capable of independent phase separation, which substantiates our hypotheses and the prediction of PSPire. This *in vitro* method offers a more controlled setting to observe the inherent phase separation characteristics of the protein, isolated from the complexities present in cellular environments.

Figure R10. Confocal images showing RAB31 droplets formed with 15%, 10%, and 5% PEG 8000 in a 20 mM Tris-HCl (pH 7.5) and 150 mM NaCl buffer.

14. For the in vitro validation, given the prediction of specific features important for the property of the given protein a direct test of the prediction is important. For example, if the interaction is predicted to be electrostatic a phase diagram using ionic strength as one variable is essential. An alternative would be to test mutants at key feature sites and show loss of ability to phase separate. Again, the goal of the validation is not necessarily whether the protein assembles but rather that the predicted features used to identify them as a hit are important for the property.

Response: According to the reviewer's comment, we conducted additional experiments to assess the impact of ionic strength on the phase behavior of PSP candidates by varying NaCl concentrations (Fig. R11). Here, SERPINB4 and PGM1 were selected for validation as these two proteins have high counts of charged stickers (10 and 11, respectively) which indicated that they were postulated to form condensates driven by electrostatic interactions. The results showed that SERPINB4 and PGM1 can form droplets *in vitro* in a buffer

containing 100 mM NaCl. At a higher NaCl concentration (300 mM), there was a marked decrease in both the size and number of droplets. At even higher NaCl concentrations (500 mM and 1000 mM), protein condensation was negligible. These observations affirmed the role of electrostatic interactions in the phase separation of SERPINB4 and PGM1. We added the results as Supplementary Fig. 6j in the revised manuscript.

Figure R11 (updated Supplementary Figure 6j). The impact of varying concentrations of NaCl on the phase separation of two PSP candidates. Confocal images of 50 μ M SERPINB4 and PGM1 in the addition of series concentrations of NaCl in buffer containing 50 mM Tris-HCl pH 7.5, 15% PEG 8000. Scale bars, 5 μ m.

15. The authors should also test proteins that are predicted to no phase separate – negative controls are really important and will provide further confirmation that the model picks up features that drive phase separation properties.

Response: Thanks for the insightful comment provided by the reviewer. According to the reviewer's suggestion, we selected ZNF738, GTF2A2, and CA5B, which were predicted as non-PSPs, for experimental validation. We generated GFP-tagged constructs of these proteins and expressed them in HeLa cells. Immunostaining images showed diffuse distribution for these proteins, supporting their lack of phase separation propensity (Fig. R12). The immunostaining images of these predicted non-PSPs were added in the updated Fig. 5b.

Figure R12 (updated Figure 5b). Validation of negative candidates. Confocal images of GFP-ZNF738, GFP-GTF2A2, and GFP-CA5B expressing in HeLa cells. Scale bars, 5 μm.

Minor comments:

1. In the first analysis to discriminate noID-PSPs from ID-PSPs and No-PSPs the supportive data provided is missing key information to interpret the conclusions. For example, it is missing False positive and False negative scores.

Response: According to the reviewer's suggestion, we added false positive rate (FPR) and false negative rate (FNR) of PSPire and other models, as shown in Table R3. The results were added to the updated Supplementary Table 2.

Table R3. Evaluation of PSPire and current predictors on six datasets.

PSPs	Dataset	Type	PSPire	SaPS	PdPS	PSPredictor	PSAP	FuzDrop	PSPer	PScore	catGRANULE	PLAAC
noID-PSPs	Testing dataset	FPR	0.21	0.36	0.37	0.78	0.40	0.80	0.58	0.77	0.46	0.50
		FNR	0.27	0.42	0.36	0.39	0.39	0.45	0.58	0.38	0.48	0.63
	G3BP1 proximity labelling	FPR	0.17	0.21	0.29	0.89	0.35	0.78	0.63	0.78	0.41	0.70
		FNR	0.10	0.20	0.22	0.23	0.35	0.44	0.58	0.32	0.28	0.47
	DACT1-particulate proteome	FPR	0.11	0.26	0.30	0.49	0.38	0.78	0.54	0.72	0.47	0.65
		FNR	0.16	0.24	0.20	0.77	0.28	0.41	0.52	0.41	0.38	0.60
	RNAgranuleDB	FPR	0.22	0.27	0.32	0.53	0.45	0.76	0.57	0.78	0.47	0.49
		FNR	0.14	0.49	0.39	0.67	0.29	0.39	0.55	0.29	0.41	0.65
	PhaSepDB_MLO	FPR	0.28	0.31	0.37	0.73	0.44	0.81	0.70	0.76	0.52	0.68
		FNR	0.25	0.53	0.39	0.53	0.42	0.41	0.41	0.44	0.48	0.58
	DrLLPS_MLO	FPR	0.21	0.31	0.37	0.77	0.46	0.80	0.71	0.79	0.56	0.78
		FNR	0.23	0.46	0.34	0.52	0.30	0.40	0.41	0.35	0.39	0.40
ID-PSPs	Testing dataset	FPR	0.18	0.24	0.19	0.35	0.33	0.35	0.37	0.31	0.37	0.34
		FNR	0.23	0.23	0.25	0.32	0.25	0.34	0.44	0.35	0.31	0.34
	G3BP1 proximity labelling	FPR	0.16	0.20	0.27	0.47	0.23	0.52	0.40	0.49	0.31	0.34
		FNR	0.14	0.20	0.17	0.29	0.31	0.38	0.44	0.40	0.31	0.53
	DACT1-particulate proteome	FPR	0.17	0.23	0.19	0.47	0.25	0.50	0.38	0.52	0.39	0.44
		FNR	0.23	0.23	0.24	0.37	0.28	0.35	0.39	0.36	0.27	0.56
	RNAgranuleDB	FPR	0.21	0.25	0.22	0.32	0.31	0.41	0.38	0.39	0.32	0.31
		FNR	0.27	0.23	0.30	0.33	0.26	0.28	0.35	0.25	0.24	0.32
	PhaSepDB_MLO	FPR	0.29	0.26	0.33	0.42	0.41	0.45	0.40	0.46	0.40	0.44
		FNR	0.39	0.40	0.32	0.37	0.32	0.34	0.44	0.37	0.32	0.44
	DrLLPS_MLO	FPR	0.28	0.27	0.33	0.44	0.35	0.46	0.43	0.46	0.40	0.44
		FNR	0.35	0.37	0.28	0.35	0.34	0.36	0.46	0.37	0.29	0.43

* *The authors should also provide the table that includes the details of tools for each machine learning algorithm.*

Response: According to the reviewer’s suggestion, we added the details of tools for each machine learning algorithm, as shown in Table R4. The information was added to the updated Supplementary Table 1.

Table R4. Details of current PSP predictors used for comparison with PSPire.

Predictor	Description	Information used for prediction	Availability	Year	Ref.
PhaSePred	Metapredictor for self-assembling proteins and partner-dependent proteins	Multiple PS-related features as well as the prediction scores from several PSP predictors	predict.phase.p.pro	2022	26
PSPredictor	Sequence-based prediction tool	Protein sequence embedding	www.pkumd.l.cn/PSPredictor	2022	25
PSAP	Machine-learning classifier based on amino acid content	Amino acid features	github.com/Guido497/phase-separation	2021	24
FuzDrop	Prediction of protein droplet-promoting propensity	Variables based on sequence	doi.org/10.1073/pnas.2007670117	2020	23
PSPer	Prediction of prion-like RNA-binding proteins	Residue-level and domain-level characteristics derived from sequence	www.bio2byte.be/b2btools/psp/elifesciences	2019	18
PScore	Predictor of proteins with pi-pi interactions	Pi-pi contact frequencies	.org/articles/31486	2018	17
catGRANULE	Prediction of granule-formation propensity	RNA binding and disordered propensities, amino acid patterns, sequence length	www.tartagli.alab.com	2016	16
PLAAC	Prediction of proteins with prion-like amino acid composition	Amino acid frequencies in prion-like domain of S.cerevisiae	plaac.wi.mit.edu	2014	15

2. Structured superficial region (SSUP) of PSPs: significant role in multivalency involved in phase separation. SSUP excludes IDRs, features from SSUP are IDR independent.

Response: We fully agree with the reviewer’s comment.

3. Please include a figure that describes the sticker and spacer model for different types of protein-protein interactions that dictate phase separation. This helps explain what the overall goal of the manuscript is.

Response: We appreciate the reviewer’s constructive suggestion. The theoretical framework, known as the stickers-and-spacers model, describes the molecular grammar underlying various phase-separating systems, which can be categorized into three distinct types: folded proteins, intrinsically disordered proteins, and linear multivalent proteins¹⁸. The computed features related to stickers, IDRs, and SSUP were designed to capture the distinct properties of the three types of stickers accurately. We added a figure to describe the different types of stickers and spacers, as shown in Fig. R13. The figure is presented as the updated Supplementary Fig. 7.

Figure R13 (updated Supplementary Figure 7). Schematic view of different types of stickers and spacers. In folded proteins, stickers refer to interaction patches on the protein surface, whereas spacers comprise regions that do not participate in these interactions. For intrinsically disordered proteins, stickers can be individual amino acids, short linear motifs, or a combination of both, with spacers being the non-interacting residues interspersed among them. In the case of linear multivalent proteins, stickers are the binding sites on the surface of multiple folded domains, while spacers are the disordered linker regions and the surface residues not involved in binding. To accurately reflect the properties of different sticker types, sticker-related, IDR-related, and SSUP-related features were calculated for each of the three protein categories individually.

4. What SSUP features differentiate ID-PSPs and noID-PSPs?

Response: It should be clarified that the existence of IDR is the only feature to distinguish ID-PSPs and noID-PSPs. All other features introduced in this study were either used to distinguish ID-PSPs and non-PSPs, or to distinguish noID-PSPs and non-PSPs. Nevertheless, according to the reviewer's comment, we examined the SSUP features between ID-PSPs and noID-PSPs. As expected, most of the SSUP features cannot distinguish ID-PSPs and noID-PSPs.

5. It would be really useful to include a diagram that illustrates the overall method and approach.

Response: We appreciate the reviewer's valuable suggestion. We added a figure to illustrate the overall method and approach of PSPire, as shown in Fig. R14. The figure is presented as the updated Supplementary Fig. 3.

Figure R14 (updated Supplementary Figure 3). Overall framework of PSPire. Two sources of phase-separating proteins (PSPs) were collected: (1) PSPs employed in the development of PhaSePred, which contained 155 PSPs and 8,801 non-PSPs for training, and 117 PSPs and 2,200 non-PSPs for testing; (2) 266 PSPs extracted from LLPSDB, PhaSePro, PhaSepDB, and DrLLPS databases. Proteins with a sequence length ≤ 100 or $\geq 2,700$ amino acids were excluded. Additionally, to enable better comparison, proteins that could not be predicted by current predictors were also filtered out. The remaining proteins were then split into training and testing datasets. The training dataset comprised 259 PSPs (195 ID-PSPs and 64 noID-PSPs) and 8,323 non-PSPs, while the testing dataset consisted of 258 PSPs (194 ID-PSPs and 64 noID-PSPs) and 1,961 non-PSPs. Subsequently, IDR- and SSUP-related features were extracted for proteins in both datasets. During the training stage, the features of proteins in training dataset were utilized for model training. Meanwhile, hyperparameters for the XGBoost classifier were optimized using Optuna. The finalized, optimized model was then employed to predict the phase separation (PS) scores of proteins in the testing dataset, followed by a comprehensive performance evaluation.

6. In Figure 4c, PdPs from PhasePred is yielding a more significant score for a few datasets in comparison to PSPire. This needs to be explained why PhasePred (PdPs) is better than PSPire in the MLO dataset.

Response: A notable highlight of PSPire is its exceptional predictive capability for noID-PSPs, which is unprecedented in the field. As to ID-PSPs, the updated version of PSPire achieved the best performance in most datasets. However, as shown in Table R5 (the updated Fig. 4c), PhaSePred achieved the best ROC in two datasets, and the best PRC in one dataset. PhaSePred is a metapredictor that incorporates multiple PS-related features as

well as the prediction scores from several PSP predictors, such as PLAAC, PScore, and catGRANULE. This integration of multimodal features yields robust predictive performance for ID-PSPs. The detailed information of PhaSePred was added in the updated Supplementary Table 1.

Table R5. AUCs of ROC and PRC for PSPire and current predictors on five human MLO datasets.

PSPs	Dataset	Type	PSPire	SaPS	PdPS	PSPredictor	PSAP	FuzDrop	PSPer	PScore	catGRANULE	PLAAC
noID-PSPs	G3BP1 proximity labelling	ROC	0.93	0.81	0.81	0.28	0.69	0.25	0.34	0.31	0.66	0.35
		PRC	0.66	0.14	0.18	0.03	0.08	0.03	0.04	0.03	0.09	0.04
	DACT1-particulate proteome	ROC	0.93	0.77	0.81	0.28	0.71	0.30	0.43	0.35	0.58	0.31
		PRC	0.60	0.14	0.18	0.04	0.11	0.04	0.05	0.04	0.08	0.04
	RNAgranuleDB	ROC	0.90	0.58	0.68	0.34	0.66	0.33	0.41	0.34	0.55	0.37
		PRC	0.28	0.04	0.08	0.02	0.04	0.02	0.02	0.02	0.03	0.02
	PhaSepDB_MLO	ROC	0.80	0.59	0.65	0.29	0.59	0.29	0.39	0.31	0.49	0.31
		PRC	0.71	0.40	0.47	0.24	0.40	0.25	0.28	0.25	0.33	0.25
	DrLLPS_MLO	ROC	0.85	0.62	0.68	0.27	0.64	0.28	0.38	0.32	0.52	0.32
		PRC	0.74	0.39	0.45	0.21	0.39	0.22	0.25	0.23	0.31	0.23
ID-PSPs	G3BP1 proximity labelling	ROC	0.91	0.85	0.86	0.62	0.80	0.54	0.55	0.60	0.76	0.60
		PRC	0.58	0.37	0.41	0.10	0.30	0.07	0.13	0.20	0.25	0.14
	DACT1-particulate proteome	ROC	0.88	0.80	0.85	0.61	0.81	0.61	0.62	0.56	0.74	0.47
		PRC	0.35	0.21	0.33	0.09	0.25	0.06	0.09	0.11	0.18	0.06
	RNAgranuleDB	ROC	0.84	0.78	0.82	0.73	0.80	0.70	0.67	0.75	0.78	0.74
		PRC	0.48	0.43	0.42	0.26	0.42	0.20	0.27	0.34	0.36	0.33
	PhaSepDB_MLO	ROC	0.72	0.68	0.74	0.64	0.70	0.64	0.60	0.62	0.69	0.59
		PRC	0.79	0.76	0.80	0.69	0.76	0.67	0.68	0.69	0.74	0.68
	DrLLPS_MLO	ROC	0.75	0.69	0.76	0.63	0.72	0.63	0.57	0.63	0.71	0.60
		PRC	0.78	0.73	0.77	0.63	0.73	0.62	0.61	0.64	0.71	0.62

7. There needs to be a clear explanation/description for why no-IDPSP protein also contain the ID region but the important feature for PS formation is it's secondary structure/domain rather than ID region. This is really confusing.

Response: In this study, the IDRs were determined using a cutoff of 20 residues (see Methods for details). With this definition, noID-PSPs do not contain IDRs. It should be clarified that the important features for PSPire prediction differed between ID-PSPs and noID-PSPs (as shown in Supplementary Fig. 4). For ID-PSPs prediction, the important features included the Phos feature, IDR-related features, and SSUP-related features. For noID-PSPs prediction, the important features only included the Phos feature and SSUP-related features.

8. Alpha-fold predictions are notoriously bad for IDR and sometimes can be annotated as secondary structure. This of course will change how PSPire uses secondary structure inferences if the model is incorrectly annotated.

Response: We fully agree with the reviewer's opinion that the AlphaFold structures could potentially influence secondary structure prediction and subsequent IDR calculation. To mitigate this impact, after filtering out secondary structures, the IDRs were further refined by iteratively converting short stretches of up to 3 disordered residues among ordered regions to ordered states, and vice versa. In addition, ordered stretches of up to 10 consecutive residues were converted to IDRs if they were flanked by two IDRs of at least 20 residues. We discussed the potential influence of AlphaFold structures in the Discussion section.

References

1. Alberti, S. *et al.* A user's guide for phase separation assays with purified proteins. *Journal of molecular biology* **430**, 4806-4820 (2018).
2. Ash, P.E. *et al.* TIA1 potentiates tau phase separation and promotes generation of toxic oligomeric tau. *Proceedings of the National Academy of Sciences* **118**, e2014188118 (2021).
3. Poudyal, M. *et al.* Intermolecular interactions underlie protein/peptide phase separation irrespective of sequence and structure at crowded milieu. *Nature Communications* **14**, 6199 (2023).
4. Agarwal, A. *et al.* VAMP2 regulates phase separation of alpha-synuclein. Preprint at <https://doi.org/10.1101/2023.06.16.545277> (2023).
5. Dubey, S., Gupta, H. & Gupta, A. Autoacetylation-mediated phase separation of TIP60 is critical for its functions. Preprint at <https://doi.org/10.1101/2023.05.06.539700> (2023).
6. Gu, S., Xu, M., Chen, L., Shi, X. & Luo, S.-Z. A liquid-to-solid phase transition of Cu/Zn superoxide dismutase 1 initiated by oxidation and disease mutation. *Journal of Biological Chemistry* **299**(2023).
7. Schmidt, T. *et al.* ProteomicsDB. *Nucleic acids research* **46**, D1271-D1281 (2018).
8. Lautenbacher, L. *et al.* ProteomicsDB: toward a FAIR open-source resource for life-science research. *Nucleic Acids Research* **50**, D1541-D1552 (2022).
9. Samaras, P. *et al.* ProteomicsDB: a multi-omics and multi-organism resource for life science research. *Nucleic acids research* **48**, D1153-D1163 (2020).
10. Feng, Z., Jia, B. & Zhang, M. Liquid-liquid phase separation in biology: Specific stoichiometric molecular interactions vs promiscuous interactions mediated by disordered sequences. *Biochemistry* **60**, 2397-2406 (2021).
11. Zhang, H. *et al.* Liquid-liquid phase separation in biology: mechanisms, physiological functions and human diseases. *Science China Life Sciences* **63**, 953-985 (2020).
12. van Mierlo, G. *et al.* Predicting protein condensate formation using machine learning. *Cell Reports* **34**, 108705 (2021).
13. Saar, K.L. *et al.* Learning the molecular grammar of protein condensates from sequence determinants and embeddings. *Proceedings of the National Academy of Sciences* **118**, e2019053118 (2021).
14. Tunyasuvunakool, K. *et al.* Highly accurate protein structure prediction for the human proteome. *Nature* **596**, 590-596 (2021).
15. Newton, J.C. *et al.* Phase separation of the LINE-1 ORF1 protein is mediated by the N-terminus and coiled-coil domain. *Biophysical Journal* **120**, 2181-2191 (2021).
16. Vendruscolo, M. & Fuxreiter, M. Towards sequence-based principles for protein phase separation predictions. *Current Opinion in Chemical Biology* **75**, 102317 (2023).
17. Yeo, J.C., Wall, A.A., Luo, L. & Stow, J.L. Rab31 and APPL2 enhance FcγR-mediated phagocytosis through PI3K/Akt signaling in macrophages. *Molecular biology of the cell* **26**, 952-965 (2015).
18. Choi, J.-M., Holehouse, A.S. & Pappu, R.V. Physical principles underlying the complex biology of intracellular phase transitions. *Annual review of biophysics* **49**, 107-133 (2020).

Reviewers' Comments:

Reviewer #1:

Remarks to the Author:

In their updated manuscript, Hou et al have addressed the major concerns of this reviewer, including significant improvements of the in vitro and in vivo validation strategies of their chosen targets. This reviewer appreciates the additional efforts of the authors to address the reviewer concerns, and now recommends it for publication without further alteration.

Reviewer #2:

Remarks to the Author:

The authors have thoughtfully addressed concerns from reviewers. Given the nicely revised manuscript, I fully endorse the publication of this work in Nature Communications.

Reviewer #3:

Remarks to the Author:

We commend the authors on incorporating and addressing our comments/suggestions. The new data and analysis assuages our concerns and fills key gaps that were missing in the original submission. I have no concerns and think the manuscript is now suitable for Nat Comm.